# Ultrasensitive imaging-based sensor unlocked by differential guided-mode resonance

Zhenchao Liu[1,2,3,6], Houxin Fan[1,6], Tingbiao Guo [1] ✉, Qin Tan[1], Zhi Zhang[1], Yuwei Sun[1], Julian Evans[1], Junbo Liang[2], Ruili Zhang[2] & Sailing He [1,2,4,5] ✉

Imaging-based sensors convert physicochemical parameters of analytes into visible patterns, yet a high sensitivity remains constrained. Here, we introduce the concept of differential guided-mode resonance with thickness modulation at a tens-nanometer scale to greatly enhance the sensitivity, alleviating the sensitivity-dynamic range tradeoff. Experimental results reveal a sensitivity of up to a million-level pixels per refractive index unit (RIU), surpassing existing technologies by nearly three orders of magnitude, with a large dynamic range reconfigured by the incident angle. With the present method, a moderate value (about 100) of the Q factor suffices to make a record high sensitivity and the Figure of Merit (FOM) can reach $10^4$ RIU$^{-1}$ level. We also demonstrate a portable device, highlighting its potential for practical applications, including 2D distribution sensing. This method unlocks the potential of imaging-based sensors with both record high sensitivity and tremendous dynamic range for accurate medical diagnosis, biochemical analysis, dynamic pollution monitoring, etc.

Medical and environmental monitoring demand higher sensitivity, larger measurement range and simplicity. This trend has spurred the continuous advancement of refractive index sensor technologies, recognized as crucial analytical tools[1-7]. Compared to other sensing methods and devices such as immunofluorescence[8], colloidal gold[9], enzyme-linked immunosorbent assay (ELISA)[10,11], mass spectrometry[12], indicator titration[13], ultraviolet absorbance spectrophotometry[14], and fluorescence quantitative PCR[15,16], refractive index sensors offer comprehensive advantages including high sensitivity, rapid detection, and high-throughput measurement. Refractive index sensors based on spectral shifts or splitting require high-resolution spectrometers, making them bulky, expensive, and unsuitable for portable or real-time applications[17,18]. Their sensitivity depends on the optical resonance structure, and the resolution is limited by the spectrometer. Phase-based sensors offer high sensitivity but require complex setups and precise phase detection, limiting their portability and real-time usage[19,20]. Intensity-based sensors monitor refractive index changes by detecting reflectance/transmittance variations. While simpler in design, they have lower sensitivity and are prone to noise and environmental interference. In contrast, an imaging-based sensor directly captures spatial light distributions or color changes, eliminating the need for complex instruments and making it more suitable for real-time and portable applications.

Imaging-based sensors directly translate detected information into image patterns, offering an intuitive and convenient picture of measured data. This capability is significant in various applications,

[1]Centre for Optical and Electromagnetic Research, Enze-ZJU Joint Lab for MedEngInfo Collaborative Innovation, College of Optical Science and Engineering, Zhejiang University (ZJU), Hangzhou 310058, People's Republic of China. [2]Taizhou Institute of Medical Health and New Drug Clinical Research, Taizhou Enze Medical Center (Enze), Taizhou Hospital, Zhejiang University, Taizhou 318000, People's Republic of China. [3]Singapore University of Technology and Design (SUTD), 8 Somapah Road, Singapore 487372, Republic of Singapore. [4]National Engineering Research Center for Optical Instruments, Zhejiang University, Hangzhou 310058, People's Republic of China. [5]Department of Electromagnetic Engineering, School of Electrical Engineering, KTH Royal Institute of Technology, Stockholm SE-100 44, Sweden. [6]These authors contributed equally: Zhenchao Liu, Houxin Fan. ✉e-mail: tbguo@zju.edu.cn; sailing@kth.se

including hazardous material detection, emergency monitoring, accurate medical diagnosis, real-time non-clinical detection, and use by non-professionals. Researchers have extensively investigated imaging-based sensors[21–24], aiming to establish a direct link between human eyes and the physical world through sensor visualization. This approach has garnered significant interest in three key areas: fluorescence sensing[22,23], Raman sensing[25–28], and refractive index sensing[29–32]. Fluorescence visualization sensors require sample modification with fluorescent markers. Measuring the concentration of antigens through changes in the refractive index due to selective binding of antigen-antibody is a principle used in some refractive index sensing techniques such as Surface Plasmon Resonance (SPR)[33–39]. Existing refractive index visualization sensors are typically complex and exhibit relatively low sensitivity, significantly limiting their range of applications. Some imaging-based SPR sensors[40,41] combine imaging tools for refractive index sensing. These sensors are essentially simple intensity-based SPR sensors, where the sensitivity is limited by the inherent sensitivity of the resonance mode. Recent studies have explored imaging-based refractive index sensors through the construction of geometric metasurfaces[42–47]. These studies use gradient structures (e.g., plasmonic gradient structures[42,43,48]) for spatial mapping. However, there lack of a method to precisely control or build a small gradient profile. As a result, these approaches generally exhibit low sensitivity. The sensitivity of these sensors is usually around $10^3$ pixel/RIU level[42,43].

In this study, we introduce the concept of differential guided-mode resonance (dGMR) to address this challenge by utilizing a thickness-modulated (at a 10-nanometer scale) chip to achieve an unprecedented sensitivity with a reconfigurable dynamic range. Refractive index information can be decoded using a thickness-modulated waveguide layer excited by a surface plasmon polariton (SPP). Through a non-lithographic approach, dGMR can be implemented at a large scale for batch fabrication. The sensitivity reaches up

to the order of a million pixels per refractive index unit (990000 pixel/RIU, surpassing existing counterparts by nearly three orders of magnitude[42,43]) with an extended dynamic range, alleviating the sensitivity-dynamic range limit. In this paper, the Figure of Merit (FOM), defined as the ratio of sensitivity to the full width at half maximum (FWHM) of the resonant spectrum, can reach $10^4$ RIU$^{-1}$ level, far beyond the theoretical limit of the SPR sensor[17]. Additionally, we have developed a portable prototype based on this sensor chip, showcasing its potential for practical applications, including 2D distribution sensing.

## Results
### The concept of differential guided-mode resonance
The guided-mode resonance (GMR) structure is based on the coupling between the SPR and GMR, utilizing the Kretschmann configuration, as shown in Fig. 1a. The incident lightwave couples to the SPP through the evanescent field, further exciting the supported GMR in the waveguide layer. We can obtain the dGMR as a single sensing unit by introducing two GMRs with small resonant-condition differences (due to the small difference of thickness $t$ of the waveguide layer, as shown in Fig. 1b) through the thickness-modulation. The key point is that the resonant thickness $t$ is a continuous and monotonic function of refractive index $n$ (giving resonance at thickness $t$), enabling the sensing capability by constructing the differential formula, as shown in Fig. 1b. As the refractive index changes, the resonance occurs from one GMR structure to the other GMR structure and one can detect the refractive index change through spatial shifts of resonance stripes. Unlike conventional optical resonance-type sensors, the limiting formula in Fig. 1b reveals that the sensitivity is determined by the thickness difference independent of the $Q$ factor.

We start with the pixelated structure with thickness modulation to demonstrate the concept of differential guided-mode resonance for

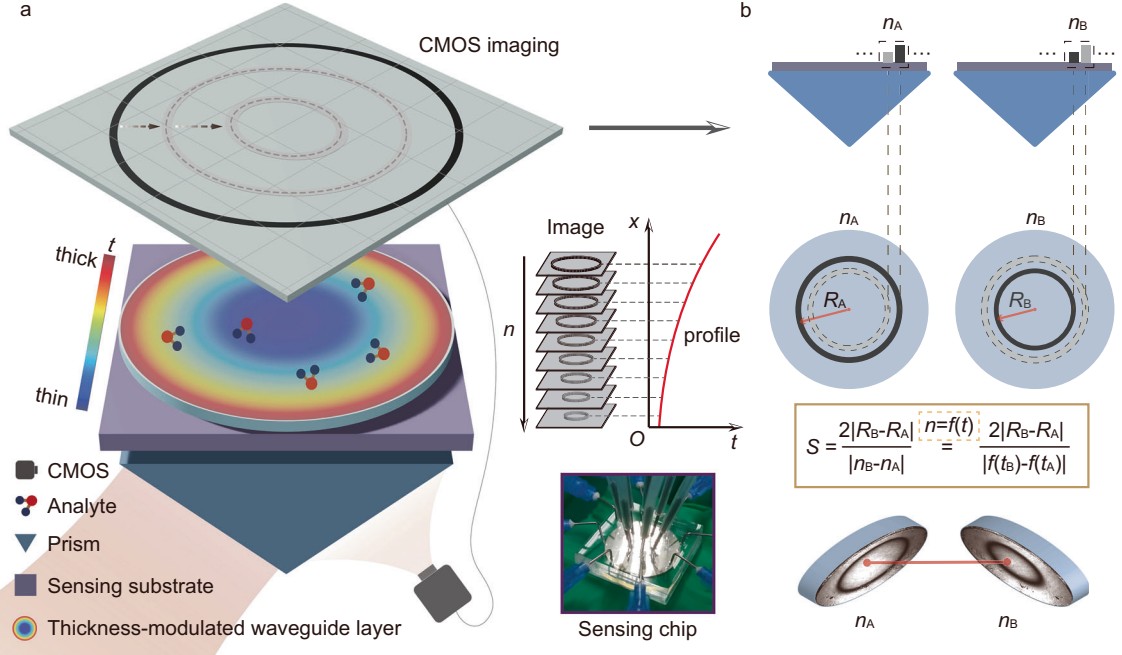

**Fig. 1 | The schematic of the ultrasensitive differential guided-mode resonance.** **a** The schematic of the thickness-modulated imaging-based sensor through the Kretschmann configuration. The thickness-modulated sensor is based on a refractive-index-sensitive and thickness-modulated waveguide layer on a metal substrate. $t$ means the thickness of the waveguide layer. The real refractive index ($n$) of the surroundings is mapped to the resonant thickness profile by modulating the thickness of the waveguide layer. The change in the image pattern can be captured by a CMOS (Complementary Metal Oxide Semiconductor) camera, which conveys

the refractive index information. **b** The proposed differential guided-mode resonance concept is illustrated with a simple two-ring structure as a waveguide layer with different thicknesses. As the surrounding refractive index $n$ changes, from $n_A$ to $n_B$, the sensing ability, which is described as the sensitivity $S$, is determined by the curve of relation between $n$ and the corresponding thickness $t$ at resonance ($n = f(t)$). The appearance of the dark ring stripe means the occurrence of the resonance. $R_B$ and $R_A$ denote the radii of resonant ring stripes of the imaging pattern, under two different refractive indices $n_A$ and $n_B$, respectively.

refractive index sensing. Figure 2a illustrates a simple model of the structure. We selected epoxy resin (SU-8) with different thicknesses as the dielectric waveguide layer and included titanium as the adhesive layer between the substrate and the metal layer. The Transfer Matrix Method (TMM) is employed for simulation, with detailed calculations outlined in Eqs. S1–S3 (Supplementary Section 1). This simple resonant structure enables incident lightwave coupling to the SPP, further exciting the GMR through coupling between SPR and GMR. The localized external electric field distribution on the surface can further improve the sensing capability through the enhancement of the electric field (Fig. S1, Supplementary Section 2).

Figure 2b(i) and (ii) illustrate the calculated SPR reflectivity angular spectrum as a function of the refractive index of the surroundings and the resonant thickness of the dielectric layer, respectively. This demonstrates that changes in either the refractive index or the thickness can lead to a significant shift in the resonant angle. Here, the refractive index of the dielectric layer is determined from measurements (Fig. S2, Supplementary Section 3). Additional simulation details are given in Supplementary Section 1.1. As depicted in Fig. 2c(i), we fabricated four dielectric patches with various thicknesses of the dielectric layer. The four patches show distinct angular spectra with the appearance of the darkest patch at each resonant angle (Fig. 2c(ii)), owing to the excitation of GMR. The $Q$ factor is calculated to be 123. Here, the thickness information was measured and fitted through incident angle scanning, indicating that this can also be used to restore the thickness profile of the thickness-modulated surface (Figs. S3 and S4, Supplementary Section 4). Figure 2d(i) and (ii) extract the relationship (labeled as $n$-$\theta$ curve) between the refractive index ($n$) and resonant angle ($\theta$) for different resonant thicknesses, and the relationship (labeled as $t$-$\theta$ curve) between the thickness ($t$) and resonant angle ($\theta$) for different refractive indices, respectively. As the thickness increases (from $t_1$ to $t_4$; Fig. 2d(i)), the $n$-$\theta$ curve shifts towards a larger resonant angle for a fixed $n$. Similarly, with increasing refractive index (from $n_1$ to $n_4$; Fig. 2d(ii)), the $t$-$\theta$ curve also shifts towards a larger resonant angle for a fixed thickness $t$. These results indicate that there exists a one-to-one mapping between the resonant thickness and the refractive index at a fixed resonant angle (vertical dark line in Fig. 2d). This relationship is shown in Fig. 2e: varying the refractive index will excite a corresponding resonant mode in the dielectric layer with a specific thickness, suggesting that the thickness-modulated dielectric layer can serve as an indicator for refractive index changes. As illustrated in the inset of Fig. 2f, the waveguide thickness is constrained by the coupling conditions (Eqs. S4–S6, Supplementary Section 5), as depicted in the cut-off area (gray area) in Fig. 2f.

### Binary image sensing with pixelated thickness-modulation

To illustrate the thickness-modulated resonance for refractive index sensing, we fabricated a thickness-modulated waveguide layer at a tens-of-nanometers scale combined with microfluidics, as depicted in Fig. 3a. This layer features quick response (QR) code-like square patches, with the thickness of each patch (randomly distributed in the thickness range from 360 nm to 430 nm) encoded through the grayscale electron beam lithography. The exposure dose matrix for all pixels is provided in Fig. S5 (Supplementary Section 6). Each patch exhibits a distinct resonant angular spectrum, and the whole sensor displays a unique gray pattern under a fixed incident angle, as illustrated in Fig. 3b. The reflectivity angular spectrum of each patch would shift along with the refractive index change, leading to a change in the overall pattern (Fig. 3b). By monitoring this pattern change, the refractive index of surroundings can be decoded effectively.

To demonstrate the ability for sensing, we measured image patterns under varying refractive index conditions with this thickness-modulated chip. As shown in Fig. 3c, the image patterns exhibit significant differences as concentrations of the glucose solution change from 0% to 10% (the relationship between the refractive index and

concentration is illustrated in Fig. S6, Supplementary Section 7). To simplify the process of extracting the refractive index, the measured images (yellow box) captured by a black and white camera were first pixelated to gray images (blue box) and then converted to binary images (gray box). From the binary images in Fig. 3c, we can clearly see the change of the pixelated pattern, as the refractive index changes. To quantitatively analyze the relationship between the refractive indices and diverse patterns, the intensity patterns under different refractive indices (or concentrations) were first calculated theoretically to establish a "binary image library" for different refractive indices (Fig. 3d(ii)). In the calculation, we used the array thicknesses obtained through the incident angle scanning (Fig. 3d(i)). The details of the angle scanning and thickness extraction are illustrated in Figs. S7–S9 (Supplementary Section 8). The binarized image patterns between the measurement and calculation are then matched to decode the refractive index. The results between the measured concentrations and the ground truth concentrations are shown in Fig. 3d(iii), demonstrating considerable accuracy (the maximum concentration error is less than 0.5% (0.00075 RIU)) across six samples (from sample A to sample F). All calculation details are outlined in Figs. S10–S14 (Supplementary Section 9). This QR code change can also inspire an optical encryption function. The refractive index or concentration information can be encoded into the QR-like pixels with different thicknesses. Upon injecting a solution of a certain concentration, the corresponding QR code is revealed as an intensity image. By scanning this QR code, the encrypted refractive index and concentration information can be decoded, as demonstrated in Fig. 3e. This method enables a blind-reading function, negating the need for precise fabrication of thickness or other structural parameters and eliminating the need for solution calibration.

This thickness-modulated array was further utilized to monitor the molecular absorption of the polydopamine (PDA) to characterize its surface sensing performance, as shown in Fig. 3f. Despite some inaccuracies, this approach allows for the decoding of refractive index information at the sensing interface and enables real-time monitoring of the molecular absorption process, as shown in Fig. 3f. Small errors may be caused by the uneven adsorption of the PDA molecules.

### Ultrasensitive sensing with a continuous and nanometer-scale thickness gradient chip

To obtain ultra-high sensitivity, we construct the dGMR with a continuous and nanometer thickness difference using a lithography-free method. In this method, the waveguide layer in the sensor chip was substituted with a thin silicon oxide ($SiO_2$) layer made from the plasma-enhanced chemical vapor deposition (PECVD). Here, UV lithography was employed to pattern the Ag layer to make a sensor array on a 4-inch wafer, as illustrated in Fig. 4a. During the $SiO_2$ deposition process with PECVD, nonuniformity always occurs, leading to a nanometer-scale gradient in thickness. The measured and calculated thickness profiles (Fig. 4a) indicate that the thickness distribution resembles a near-spherical shape, with a minimal thickness in the middle area and increasing gradually towards the edges.

After bonding the chip with a microfluidic chip, six glucose solutions with varying concentrations (ranging from 0.0 wt% to 1.0 wt%) were injected into the sensing chip. As expected, the measured images reveal distinct ring patterns under different concentrations (Fig. 4b(i)). Even a minor increase in the refractive index can lead to a significant shift of the ring pattern, moving gradually from the edge toward the center. The pixel shift as a function of concentration is extracted and shown in Fig. 4b(ii), reflecting a pronounced sensitivity. Note that Fig. 4 highlights a continuous thickness variation enabled by a lithography-free method (aligned with conceptual Fig. 1) while Figs. 2 and 3 focus on discrete thickness differences, represented by a pixelated structure. These four figures are presented in different forms, but their fundamental concept remains the same—constructing thickness difference.

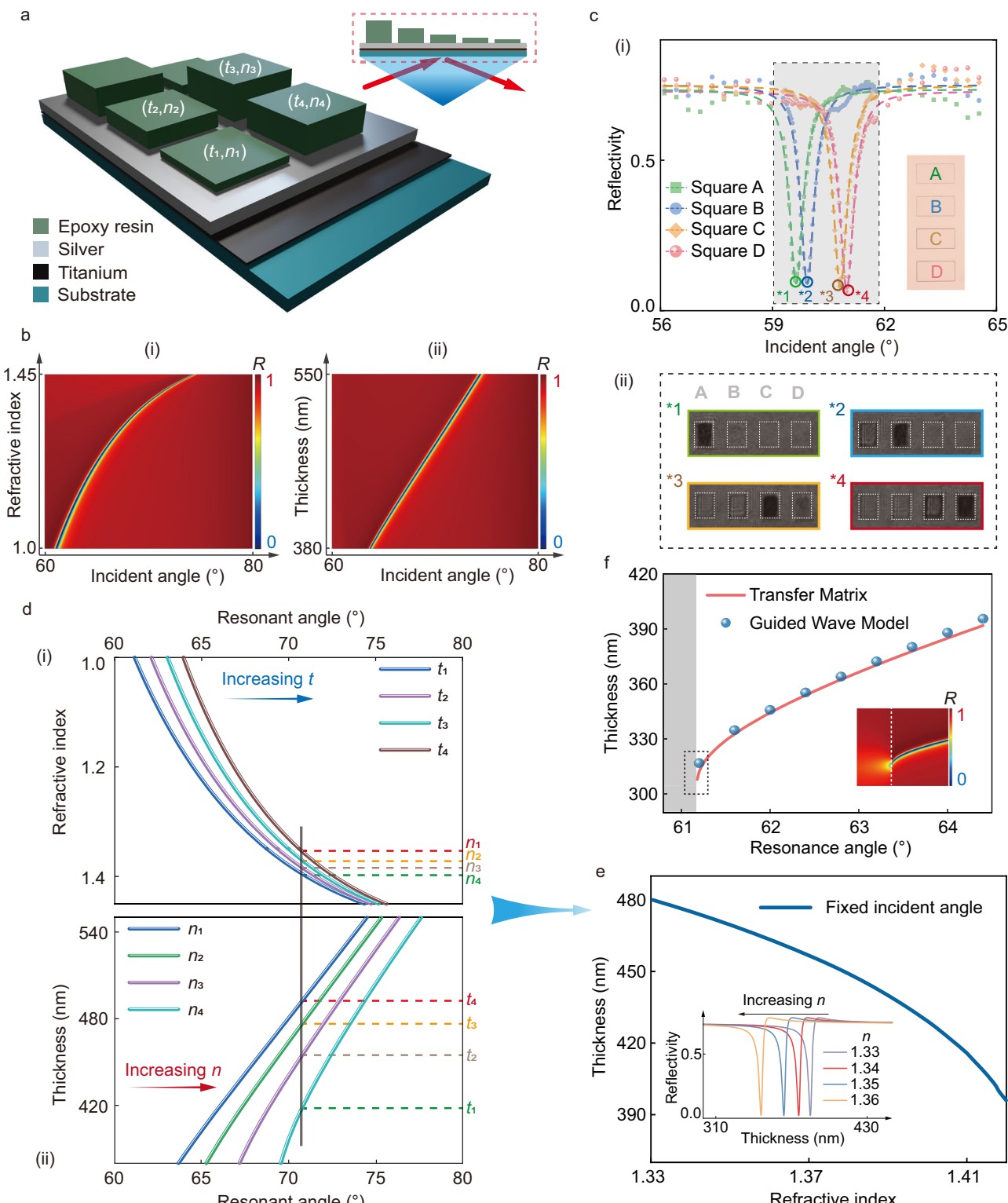

**Fig. 2 | The simulated and experimental results of thickness-modulating for refractive index sensing. a** The SU-8 multi-square resonant structure with different thicknesses through guided-mode coupling. The arrow indicates the incident lightwave. $(t,n)$ means the thickness and refractive index at resonance of SU-8 square. **b** Simulated reflectivity angular spectrum changing with the refractive index (i) and thickness (ii). $R$ in the scale bar means reflectivity. **c** The measured reflectivity angular spectrum (i) of our sensing system with a four-square resonant structure. The gray area in the figure means the resonance area. The captured gray images (ii) show a unique pattern under a corresponding incident angle. The 'A-D' letters mean different resonant structures and the asterisk number indicates the resonant dip. **d** The resonant incident angle changes with the refractive index under different thicknesses (i) and changes with the thickness under different refractive indices (ii). The dark line crosses the two figures stands for a fixed resonant angle, which corresponds to the experimental conditions (with a fixed incident angle). **e** The one-to-one relationship between the refractive index and the resonant thickness for a fixed incident angle. The inset is the reflectivity when the thickness varies at different refractive indices. **f** The relationship between the thickness and the resonant angle near the cut-off area, calculated via TMM and guided wave model (phase matching condition) for $n = 1.33$. The gray area means the cut-off area. The inset is the zoom-in near the cut-off area.

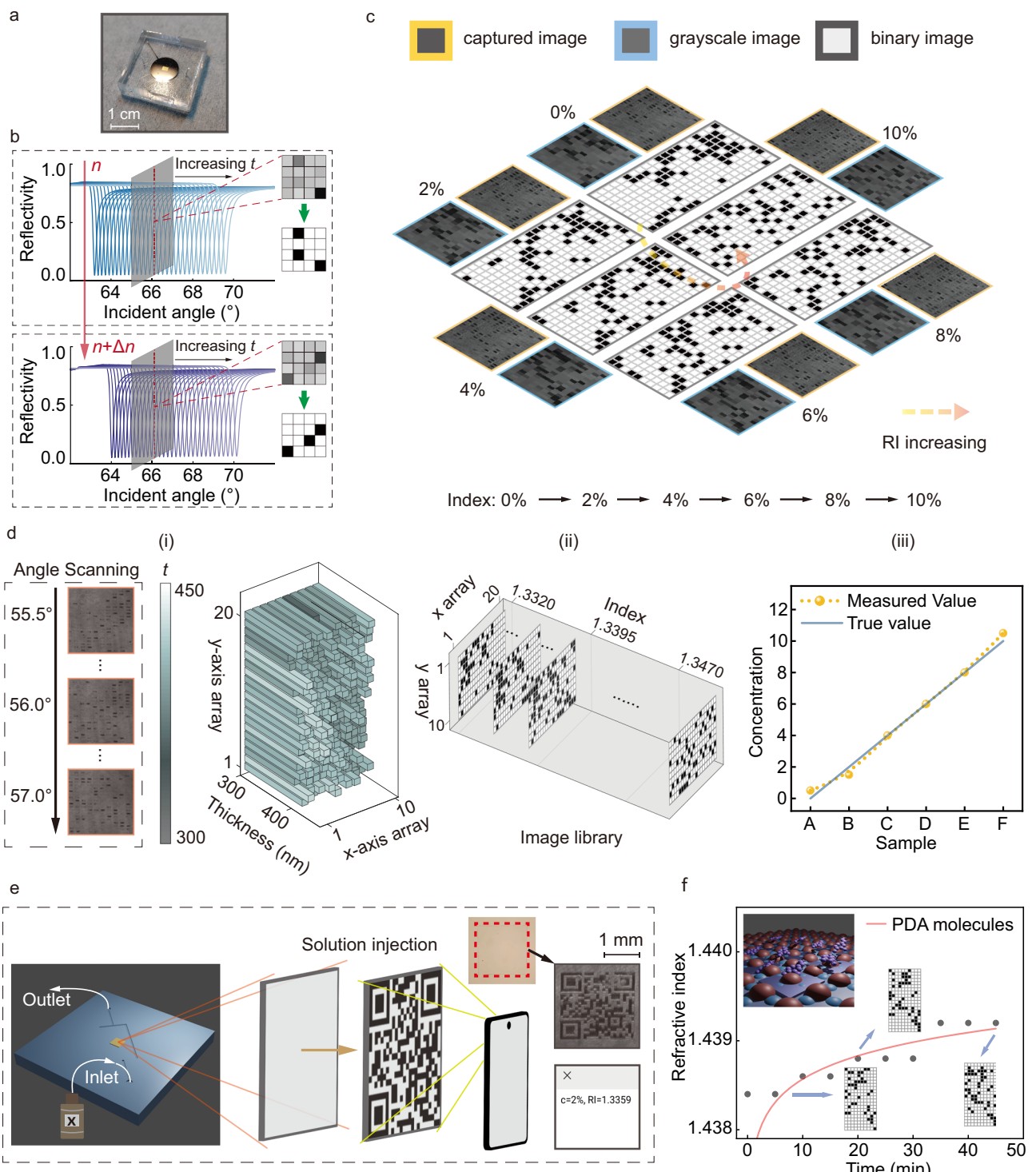

**Fig. 3 | The sensing experimental results of the thickness-modulated chip through dGMR. a** The fabricated sensing chip combined with a microfluidic system. **b** The simulated unique reflectivity patterns of the thickness-modulated resonant array under two different refractive index conditions ($n$ and $n + \Delta n$). $t$ means the thickness. **c** The measured image patterns under six different refractive indices. For each concentration of glucose solution, the yellow box represents the captured image, the blue box represents the grayscale image, and the gray box represents the binarized image. **d** Angle scanning and the calculation of the refractive index. (i) The thickness calculation based on the resonant angle scanning through TMM. (ii) The calculated binary image library for different refractive indices. (iii) The comparison between the measured concentration values and true values for samples with varying concentrations. **e** The QR code decryption application using the thickness-modulated chip, scanned by the smartphone camera. **f** The monitoring for the absorption of PDA molecules.

With a fixed incident angle, the ring stripe shrinks until it disappears completely as the concentration increases beyond 1.0%. Consequently, the dynamic concentration range for this incident angle is nearly (0.0%, 1.0%) (approximately 0.0015 RIU). To extend the dynamic concentration range, the sensor can be reset by increasing the

incident angle. As a result, the ring stripe of the 1.0% concentration sample can be re-adjusted (or repositioned) to the edge area with a larger incident angle, supported by simulation results (Fig. 2b(i) and (ii)). Thus, by simply adjusting the incident angle, the new measurable cycle can widen the dynamic concentration range in an adjustable way,

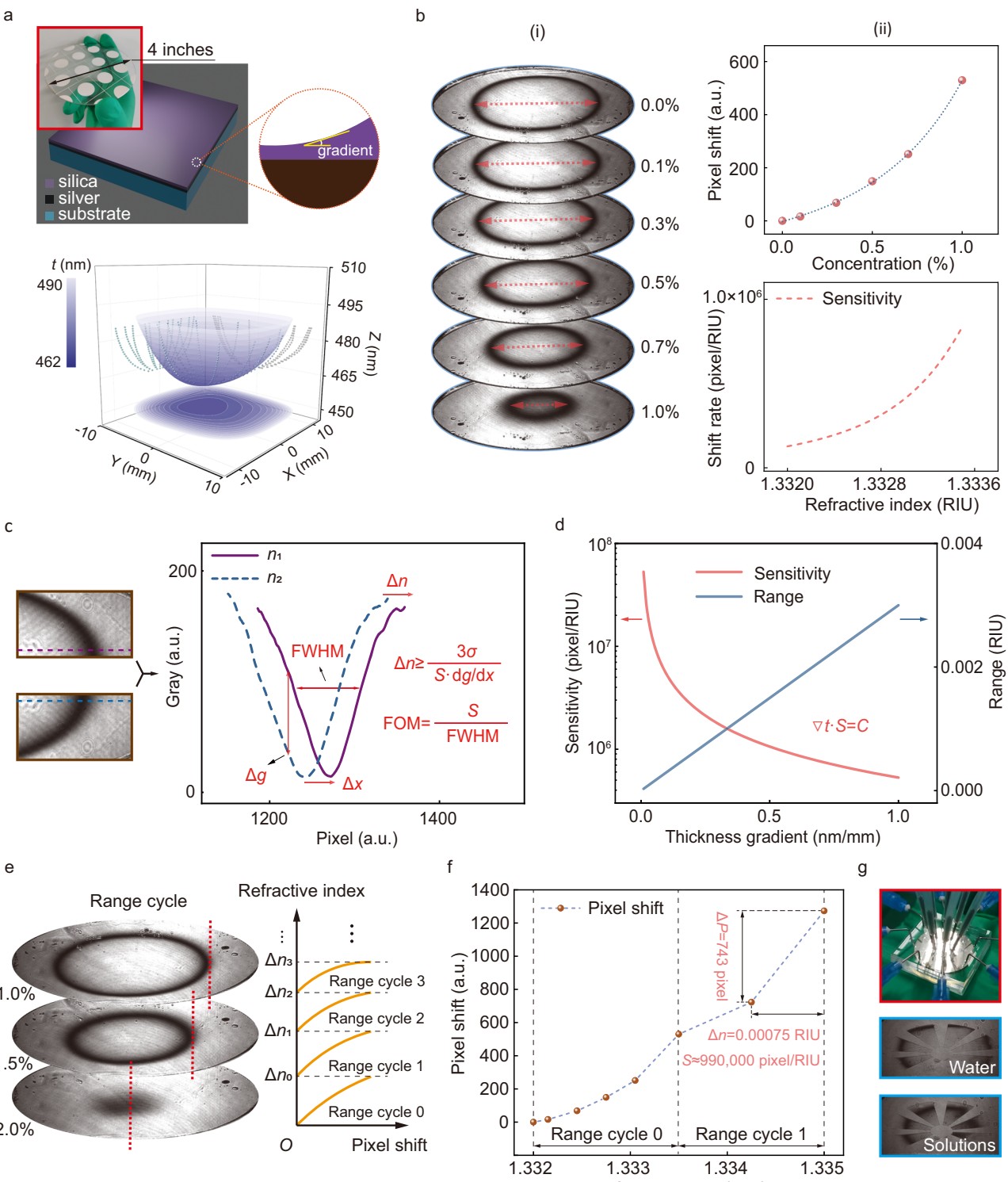

**Fig. 4 | Ultrasensitive refractive index imaging sensor based on the ring stripe shift of the thickness-modulation chip through the dGMR. a** The structure of the sensing chip and the measured and calculated thickness profiles. In the three-dimensional coordinate system (*x-y-t*), a point represents the thickness of the dielectric layer at position (*x-y*), and the spatial distribution of the thickness is characterized by the surface fitted by the discrete points. The image in the *x-y* plane is a contour map of the fitted thickness spatial distribution. **b** The refractive index sensing results. (i) The measured distinct ring pattern images under different concentrations. (ii) The curve between the pixel shift and concentration, as well as

the sensitivity curve. **c** The sensing resolution analysis. **d** The analysis of sensitivity and dynamic range as thickness gradient varies. **e** The new measurable cycle, which can widen the dynamic range of concentrations, from range (0, 1.0%) to additional range (1%, 2%), and so on, by using different incident angles in different measurement cycles. Dynamic range cycles can be continuously increased by simply re-adjusting the incident angle. **f** In the new dynamic range, the maximum experimental sensitivity can reach 990000 pixel/RIU (The resolution of the CMOS camera is 1608 × 1104 pixels). **g** The multi-channel sensing chip setup and the high-throughput imaging sensing with different solutions.

as depicted in Fig. 4e, for a new concentration range (1.0%, 2.0%) (approximately 0.0015 RIU). As shown in Fig. 4e, the dynamic range ($\Delta n = \Delta n_0 + \Delta n_1 + \Delta n_2 + \cdots$) can be expanded in multiples of the standard dynamic range ($\Delta n_0$), in a continuous angle re-adjusting process, alleviating the tradeoff between the sensitivity and dynamic range. The applicable range of refractive index for this example is approximately from 1 to 1.3675 (Fig. S15, Supplementary Section 10). In the extended dynamic range, the maximum experimental sensitivity can reach 990000 pixel/RIU, as shown in Fig. 4f, setting a record high sensitivity exceeding counterparts[42,43] by nearly three orders of magnitude. The purpose of these data points is to calculate the sensitivity in the high refractive index range, not to establish a calibration curve. Furthermore, microfluidic technology was combined to create a multi-channel sensing chip, as shown in Fig. 4g. This multi-channel sensing chip setup can achieve high-throughput imaging sensing by injecting different solution samples through the pipe.

The refractive index resolution of our chip is determined by two key parameters, as illustrated in Fig. 4c: the fineness and the pixel shift (sensitivity) of the stripe. A narrower resonant peak and a larger thickness difference can improve the fineness of the stripe, and hence the refractive index resolution. However, a larger thickness difference implies that a larger range of refractive indices is mapped in the measurement range, which can decrease the pixel shift sensitivity of the ring stripe. Additionally, the noise of the image sensor also affects the accuracy of the stripe shift. Overall, the refractive index resolution ($\Delta n$) can be expressed as a function of the gradient of the intensity over position ($dg/dx$), the sensitivity ($S$), and the noise of the image sensor ($\sigma$), as shown in the inset of Fig. 4c. The calculations are shown in the Figures S16–S19 (Supplementary Section 11). Based on the measured sensitivity and FWHM of stripes, the FOM reaches $10^4$ RIU$^{-1}$ level (Fig. S20, Supplementary Section 12), far beyond the theoretical limit of SPR sensor[17]. As the gradient of thickness ($\nabla t$) decreases, the sensitivity will increase as an inverse proportional function, as shown in Fig. 4d. The dynamic range for a certain incident angle is nearly proportional to the gradient of thickness ($\nabla t$). The details of the calculation are explained in the Supplementary Section 11. Besides, the repeated measurement error is analyzed in the Figure S21 (Supplementary Section 13) and the details on the method to determine the pixel shift of the ring stripe are shown in the Supplementary Section 14.

The high-specificity binding between streptavidin and biotin is a well-established model in biomolecular recognition, playing a crucial role in medical diagnostics, food safety, and environmental monitoring. As shown in Fig. 5a, we functionalized the sensing surface by silanization and streptavidin incubation to enable biotin detection. The detailed modification steps are provided in the Supplementary Section 15. Notably, even at a low biotin concentration of 1 nmol/L, our sensor exhibits a signal response exceeding 20 pixels, demonstrating a good real-time sensing performance for monitoring molecular binding at the surface.

A portable prototype ($20 \times 14 \times 8$ cm$^3$) was also investigated based on a plug-and-play sensor chip, as illustrated in Fig. 5b(i). The detector for this device was replaced by a smartphone. In Fig. 5b(ii), different ring-stripe images under various solution conditions, including air and other solutions, are captured with a smartphone camera. A clear shift of the ring stripe is observed as the refractive index changes. Unlike single-point humidity sensors, the proposed imaging sensor can support two-dimensional humidity sensing as illustrated in Fig. 5c(i). Areas with higher humidity absorb more water molecules on the sensing surface, altering the surface refractive index and consequently changing the ring stripe in these areas (Fig. 5c(i)). As a proof-of-concept demonstration, Fig. 5c(ii) shows the ring-stripe images changing with increasing humidity (from 33.5% to 52.2%) in a chamber, captured by the smartphone camera. The imaging-based sensor in this article can support the dynamic perception for the two-dimensional distribution of the refractive index or particles, which can be obtained through incident angle scanning. This scanning method is analyzed and discussed in Figs. S26 and S27 (Supplementary Section 16).

## Discussion

The present method is based on a hybrid resonance sensing chip, which can be optimized further in the future[49]. Leveraging advanced nanofabrication techniques, such as 3D printing and lithography-based methods, for large-scale or array-based precise thickness control represents a promising research direction[50–55]. Downsizing is crucial for future integrated applications, aiming for widespread use in imaging, readability, and ultrasensitive refractive index sensors, including integration with smartphones and their analysis software. Moreover, we also include a detailed discussion on the sensitivity of the imaging sensor to variations in the incident angle and the collimation of the incident waves (Supplementary Section 17).

In conclusion, we have introduced a concept of differential guided-mode resonance to revolutionize the effort for ultimate sensitivity by thickness modulation at tens-nanometer scale in a planar waveguide structure, achieving an unprecedented sensitivity of nearly one million pixel/RIU (990000 pixel/RIU) with an extended and wide dynamic range, nearly a three-order improvement over existing counterparts[42,43]. Moreover, we also include a simple table for the detailed comparison in the Supplementary Section 18. Such large sensitivity is owing to the slight difference in the dGMR structure (Supplementary Section 19). Unlike traditional approaches that rely on a high $Q$ factor, our method significantly enhances sensitivity with minimal dependence on the $Q$ factor. Moreover, we also verified the reproducibility of PECVD-based fabrication and included a more comprehensive analysis, which is shown in the Supplementary Section 20. The detailed fabrication process and the precise thickness control through dose modulation are also discussed in the Supplementary Sections 21 and 22. Additionally, by simply re-adjusting the incident angle, the dynamic range can be re-adjusted and widened, alleviating the tradeoff between the sensitivity and dynamic range. The FOM in this paper can reach $10^4$ RIU$^{-1}$ level, far beyond the theoretical limit of SPR sensor[17]. We have successfully developed a portable prototype based on large-scale, batch fabrication of sensing chips, showcasing its potential for practical applications. Our work introduces a concept for achieving ultimate sensitivity, establishing a benchmark in both record-high sensitivity and large dynamic range. It will advance imaging or readable-based sensors and promote point-of-care testing, biochemical analysis, pollution monitoring, etc.

## Methods

### Numerical simulation

The transfer matrix method (TMM) calculation for multi-layer resonant structure at 10 nanometer scale is simulated by MATLAB. For Figs. 2 and 3, the waveguide layer is made up of the SU-8 1030 photoresist. The refractive index of the SU-8 photoresist is determined by measurement. The refractive index of other materials is taken from the refractive index library. Other simulation details of Fig. 2b(i) and (ii) are: the wavelength of incident lightwave ($\lambda = 671$ nm), 1st layer (prism layer/substrate layer, $\varepsilon_1 = 2.3043$), 2nd layer (titanium adhesive layer, $\varepsilon_2 = -7.2269 + i21.580$), 3rd layer (silver layer, $\varepsilon_3 = -20.917 + i0.43400$), 4th layer (SU-8 waveguide layer, $n = 1.5995$, $t = 450$ nm), 5th layer (sensing medium layer, $n$ ranges from 1 to 1.45 in Fig. 2b(i), $n = 1.33$ in Fig. 2b(ii)). In Fig. 4, the waveguide layer is made up of silica. The refractive index of silica is determined by measurement. The construction of the "binary image library" and image matching process are achieved by the homemade MATLAB scripts. The guided wave model in Fig. 2 is calculated through MATLAB, based on the phase matching condition. The localized electric field distribution is calculated using FDTD software. The materials parameters are adopted from the built-in materials library. The recovery of the thickness profile in Figs. 2–4 is achieved with homemade MATLAB scripts based on the TMM model.

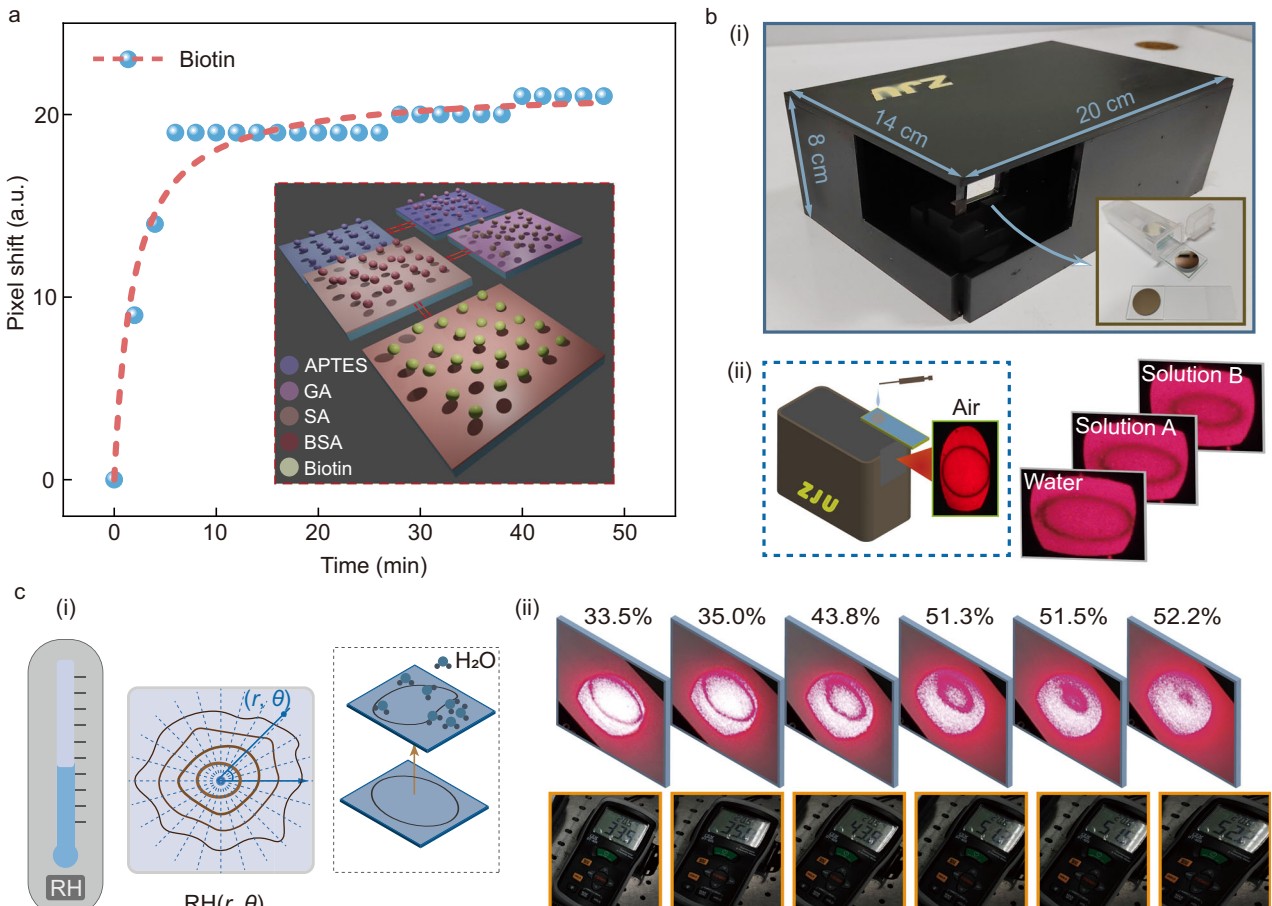

**Fig. 5 | The surface sensing and monitoring and the portable device designed for this thickness-modulated refractive index sensor. a** The monitoring response curve of the biotin molecule binding event (the concentration is 1 nmol/ L). The insert is the surface modification process. APTES means 3-aminopropyltriethoxysilane. GA means glutaraldehyde. SA means streptavidin. BSA means bovine serum albumin. **b** The portable device design and its application. (i) A portable prototype ($20 \times 14 \times 8$ cm$^3$) based on a plug-and-play sensor chip. (ii) Different ring-stripe images under various solution conditions. **c** (i) Two-dimension humidity ($RH(r, \theta)$) sensing based on this thickness-modulated chip. (ii) The ring-stripe images as the humidity increases in a chamber, captured by the smartphone camera.

## Design and fabrication

For Figs. 2 and 3, the fabrication process of the four-patch chip and the code-like sensing chip includes photolithography, metal evaporation, lift-off process, electron beam lithography and microfluidic chip sealing. The photolithography process includes spin coating, pre-baking, exposure, and development. AZ5214 photoresist is used for lithography with a mask aligner machine (SUSS Micro Tec, MA6). Titanium and silver are evaporated (Denton, Explorer) with thicknesses of approximately 2 nm and 50 nm, respectively, supporting the SPP mode. After that, the lift-off process is carried out, using acetone and ultrasound to remove the photoresist. In the electron beam exposure process (Raith 150 TWO), we use SU-8 1030 as the electron beam resin layer, whose thickness can be affected by the exposure dose. The microfluidic chip sealing is achieved with the plasma cleaner (Harrick, PDC-002) through the covalent bonding method. In Fig. 3, the "QR" patches (waveguide layer) with different thicknesses are produced by controlling the exposure dose in the electron beam exposure process. In Fig. 4, the fabrication process for patterning the Ag layer to make a sensor array on a 4-inch wafer includes photolithography, metal evaporation, lift-off process, silica deposition and microfluidic chip sealing. The silica is deposited by the PECVD (made by Surface Technology Systems Ltd., model M/PLEX CVD), with the thickness difference introduced through the deposition error. The multi-channel sensing chip is achieved through the array-type microfluidic chip technology based on the silicon substrate mold fabricated by the lithography process. The optic mounts of the prototype in Fig. 5 are designed and fabricated with customized 3D printing technology. The humidity sensing box in Fig. 5 is based on a homemade air chamber, which is built using an acrylic board.

## Measurement and characterization

In Figs. 2 and 3, for the angle scanning process, we use a programmable rotation platform. The rotation platform rotates step-by-step while the reflective images are captured by a CMOS camera (ZWO, ASI432MM). The pair of N-BK7 prisms (LBTEK) makes the direction of reflective light nearly unchanged during the angle scanning process. The optical microscope images in Figs. 2 and 3 are obtained with the Olympus BX53M microscope. For the standard thickness measurement of the four patches in Fig. 2, we use the thickness meter (Filmetrics F40-UV). The QR code in Fig. 3 is generated using the QR code generation website. We scan the QR code in Fig. 3 with a smartphone. The light source used in this system is a 671 nm laser (MRL-III-671-100mW). The characterization of the refractive index of the SU-8 epoxy resin and silica layer is achieved by an ellipsometer (HORIBA, UVISEL). In Fig. 3, solutions with different concentrations (from 0% to 10%) are sequentially injected into the microfluidic channel and come into contact with the surface of the sensing chip. After each injection process, the reflected images are captured by a CMOS camera. In Fig. 3, the polydopamine (PDA) tris solution (pH=8.5) is injected into the microfluidic channel, with the reflected image recorded every five

minutes to monitor the molecule absorption process on the surface of the sensing chip. In Fig. 4, the angle scanning process is achieved through a programmable rotation platform with a small step. The thickness profile of the silica waveguide layer is fitted based on the TMM model. In Fig. 4, solutions with different concentrations (from 0.0% to 1.0%) are sequentially injected into the microfluidic channel and come into contact with the surface of the sensing chip. After each injection, the reflected images are captured through a CMOS camera to record the position of the ring stripe. In the multi-channel sensing test in Fig. 4, the solutions with different concentrations are injected into different channels. The humidity testing in Fig. 5 is performed in an air chamber. The change in humidity is produced by a humidifier. During this testing, the standard humidity is characterized at the same time with a commercial hygrometer (CEM, DT-625).

## Data availability

The data that support the findings of this study are available from the corresponding authors upon request. Source data are provided with this paper.

## Code availability

The codes for data analysis are available from the corresponding author upon request.

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

## Acknowledgements
S.H. acknowledges the support from the "Pioneer and Leading Goose" R&D Program of Zhejiang (Nos. 2025C02159, 2025C02140, 2023C03083, 2024C03045), the National Natural Science Foundation of China (W2412107 and 91233208), the National Key Research and Development Program of China (2022YFB2804100), Ningbo Science and Technology Project (Nos. 2023Z179, 2023Z122, 2024Z146), Science and Technology Plan Key Project of Taizhou City (24gyz01), Ningbo Public Welfare Research Program Project (2024Z234), the Special Development Fund of Shanghai Zhangjiang Science City and the Special Development Fund of Hangzhou Chengxi Sci-tech Innovation Corridor. T.G. acknowledges the support from the National Natural Science Foundation of China (62105284), the National Key Research and Development Program of China (2022YFC3601002), and the Fundamental Research Funds for the Central Universities (226-2024-00150). We thank the core facilities and cleanroom provided by the Center for Optical and Electromagnetic Research in the College of Optical Science and Engineering, Zhejiang University.

## Author contributions
Z.C.L. and H.X.F. fabricated the samples and conducted the characterization while Z.C.L. and T.B.G. processed the data. Z.C.L. wrote the first draft. T.B.G., Q.T., H.X.F., Z.Z., Y.W.S. and J.E. contributed to discussions and revisions. J.B.L. and R.L.Z. contributed to discussions and funding. S.H. supervised the project, led the discussions, and finalized the manuscript. All authors have gone through the manuscript.

## Funding

## Competing interests
The authors declare no competing interests.
