## [Transparent Peer Review file · Nature Communications]

Ultrasensitive Imaging-based Sensor Unlocked by Differential Guided-Mode Resonance

Corresponding Author: Professor Sailing He

Version 0:

Reviewer comments:

Reviewer #1

(Remarks to the Author)

This manuscript introduces a refractive index imaging sensor utilizing SPP structures of varying thicknesses. The differential guided-mode resonance, achieved through precise adjustment of structure thickness, results in high sensitivity to changes in the surrounding refractive index. Additionally, the authors showcase the potential application of this method in portable devices and propose an alternative low-cost fabrication process. This work is of significant interest to the research community focused on image sensors and SPP devices. Therefore, I support the publication of this manuscript, provided my concerns are addressed.

My primary concern is regarding the manuscript's claim of "without sensitivity-range tradeoff." With a limited number of CCD pixels, there is always a tradeoff between sensitivity and the range of the proposed imaging sensor. While the measurement range of the sensor can be adjusted by the incident angle, as demonstrated in the manuscript, this constitutes a reconfiguration of the sensor rather than an elimination of the sensitivity-range tradeoff. The incident angle should be considered as part of the sensor's configuration. For instance, in a grating-based spectrometer with limited CCD pixels, finer spectral resolutions can be achieved using gratings with larger linear dispersions, but the CCD size limits the measured spectral range. Similarly, the measured spectral range can be extended by varying the incident angle, which is a reconfiguration of the spectrometer.

The schematic diagram (Figure 1) is somewhat confusing. This diagram depicts a structure with concentric rings, which aligns with the lithography-free method shown in Figure 4. However, Figure 2 depicts a pixelated structure, making the explanation of the device design rather unclear.

Additionally, please provide more details on the sensitivity of the imaging sensor to variations in the incident angle and the collimation of the incident waves.

Reviewer #2

(Remarks to the Author)

In this manuscript, the authors introduced an innovative, imaging-based refractive index sensor employing differential guided-mode resonance (dGMR) approach to achieve ultra-high sensitivity without the typical tradeoff between sensitivity and dynamic range. By modulating the thickness of the waveguide layer at the nanometer scale, this sensor translates refractive index changes into highly sensitive imaging patterns, reaching sensitivities of up to million-level pixels per refractive index unit. This result represents a significant advancement over previous studies. I recommend that this manuscript be considered for publication following moderate revisions. Below, I have outlined several points for the authors to address:

1. Metasurface-based refractive index sensors are a well-explored area of study. To provide a suitable benchmark, the authors should briefly compare image-based sensors and other types of refractive index sensors in the introduction, such as those based on spectrum shifting, spectrum splitting, phase changes, and efficiency changes. This comparison would help contextualize the performance and advantages of the proposed sensor within the broader field.
2. The manuscript does not clearly explain the physical mechanism behind the remarkable sensitivity of "990,000 pixels/RIU, surpassing existing counterparts by nearly three orders of magnitude." It is unclear if this sensitivity is primarily due to the unique differential guided-mode resonance (dGMR) mode. A more detailed comparison with existing image-based sensors, particularly references 34 and 35, would strengthen the paper. Additionally, given that the detector's pixel

- density may significantly impact this value, the authors should also discuss this factor to provide a more rigorous result.
3. To expand the refractive index detection range, the authors introduced the incident angle as a new degree of freedom. However, if the structural dimensions were sufficiently large, for example, 8 inches instead of 4 inches as mentioned in the manuscript, could it be possible to cover a wide refractive index range without altering the incident angle? The authors should consider comparing these two approaches, especially regarding the potential errors and calibration issues associated with adjusting the incident angle. This comparison would provide a more comprehensive assessment of the method's feasibility and stability in practical applications.
 4. In Figure 4, the authors employed the PECVD method, using deposition errors to introduce thickness difference in the thin film, which is an interesting approach. However, does this imply that the fabrication uncertainty may be higher compared to lithography? How did the authors control the fabrication parameters to achieve the desired thin-film thickness gradient? Additionally, what is the fabrication repeatability of the thin-film thickness gradient in the sensors? These factors are crucial for the accuracy and reproducibility of sensing performance.
 5. Recent published papers about resonance engineering or meta-sensing can be considered to be involved in References, e.g., Nature Communications volume 15, 9658 (2024); ACS Nano, 11598-11618 (2022).
 6. In Figure 5a(ii), the pixel shift after each step appears unclear. Shouldn't the pixel shift be cumulative, reflecting the overall change in refractive index across the sensor? Additionally, the curve fitting for R-IgG seems less accurate compared to the other three graphs. The authors should provide a brief analysis of this discrepancy to clarify the potential causes and implications for the sensor's performance.

In summary, the manuscript presents an innovative design with pioneering implications for the field of imaging-based refractive index sensors. It provides valuable references and insights for advancing this area of research.

Reviewer #3

(Remarks to the Author)

The authors present a ultrasensitive imaging-based sensor with large sensitivity range using guided-mode resonance, which seems to be promising in the field of imaging-based sensors. This manuscript could be published if the authors can clarify following questions:

1. Studies have reported on target detection methods utilizing imaging-based surface plasmon resonance (SPR) (e.g., Analytical Chemistry, 2001, 73(22): 5525-5531; Lab on a Chip, 2007, 7(9): 1206-1208). Additionally, there is a body of work addressing the achievement of ultrasensitive sensing through the modulation of the geometric morphology of plasmonic sensors (e.g., Nanoscale, 2019, 11, 12471). Given the existing literature, could you elaborate on the unique innovation in your approach, particularly with respect to the regulation of height changes in the geometric morphology of the sensor?
2. From the results presented, your sensor appears to demonstrate an ultra-high sensitivity of 990,000 pixels/RIU. However, upon a thorough review of the manuscript, this value seems to correspond more closely to a theoretical calculation of sensitivity. Typically, refractive index sensitivity is calculated using the formula ΔP (pixel shift) / Δn (change in refractive index). Could you clarify why the ratio between pixel shift and refractive index in your S9 formula exhibits a two-fold relationship? Furthermore, the inclusion of height change in the sensitivity calculation is intriguing. Could you provide a more detailed rationale for incorporating this factor into the formula?
3. When comparing your work with that in the literature on direct imaging of sensors based on changes in geometric morphology (e.g., Advanced Materials, 2021, 33(29): 2100270), it appears that the sensitivity of the sensor in that study is approximately 500,000 pixels/RIU, based on the method you have applied. Could you please explain how your sensor demonstrates superior performance in comparison to this work? Specifically, what are the key advantages that make your sensor more sensitive?
4. The manuscript does not provide refractive index sensitivity values for the test solutions presented in Figure 5. Since the resonance angle is expected to vary with the refractive index gradient, the nanoarray presented, which undergoes significant height changes, does not appear to exhibit continuous variation (as shown in Supporting Information S8, S9). Could you provide 2-3 SEM images at a magnification of $\times 5000$ to $\times 10000$, as well as AFM images at tilted angles, to characterize the uniformity and height variations of the sensor's microstructure? This would enable a more accurate calculation of the actual sensitivity based on data from the fabricated structures.
5. The plot in Figure 4(f), which illustrates the relationship between pixel position and refractive index, presents only three data points. This limited data set is insufficient to provide convincing evidence of a reliable relationship. Additionally, it is unclear why two lines segments with different slopes are used; could you clarify why higher data points were selected for the refractive index calculation? It would be more robust to test at least five data points to improve the reliability of the conclusion.
6. Regarding the pixel shift calculation shown in Figure 4(b), could you explain the method used to determine the pixel shift? Was this calculated through an averaging process over the entire circle, or was a specific region analyzed? Moreover, based on the physical demonstration in Figure 4(g), it seems that the fabricated template is not a perfectly regular circle, with observable variations in the thickness and placement of the stripes across different regions. Does this method exhibit reproducibility? To strengthen the analysis, could you provide error bars for the data to support the claims and clarify any potential variations?

7. This work achieves ultrasensitivity by fabricating thickness-tunable guiding resonators that align with the SPR resonance angle. However, the reported ultra-sensitivity appears to include variations in distance due to processing errors (e.g., deposition). To allow readers to more accurately assess the sensing performance, it would be helpful if the authors provided the height interval spacing, including the effects of fabrication errors. A reasonable spacing range would better inform the evaluation of the sensor's sensitivity.

8. To the best of our knowledge, achieving precise control over the height of structures during nanofabrication remains a significant challenge. The manuscript does not clearly demonstrate how height-controllable fabrication was achieved, nor does it provide data or microstructural characterization to support the reliability and reproducibility of the proposed method. From Figures 5(iii) and (iv), the observed changes in the image with humidity appear irregular. In light of this, it raises concerns about the reliability of the sensitivity calculations based on these image changes.

9. I recommend that the authors address these concerns by providing more comprehensive details on the fabrication process, including any techniques employed to ensure height control, as well as supporting data and microstructural analysis to substantiate the reliability and reproducibility of the approach. Furthermore, a more thorough examination of the relationship between image changes and sensitivity, particularly in response to humidity variations, is necessary to validate the reported sensitivity values.

10. Based on the data presented, it remains unclear how the sensor achieves sensitivity and range without apparent trade-offs, particularly given that the system seems to only detect small refractive index changes. Could you elaborate on how the system manages to maintain high sensitivity over a broad range of refractive index variations? Furthermore, do larger refractive index changes pose any challenges to the sensor's detection capabilities?

In Figure 5(ii), the non-specific adsorption curve for R-IgG shows only a 3-pixel shift, which seems inconsistent with the claim of "ultrasensitivity" made in the manuscript. This raises concerns about the validity of the sensitivity calculation method employed. Could you provide a more thorough explanation and justification for the sensitivity calculation, especially considering the discrepancy between the expected and observed shifts?

Additionally, in light of the ultra-sensitivity claimed, does the sensor demonstrate robust performance in practical applications, such as the detection of ultra-low concentration gases or even single-molecule detection? As a reviewer, I recommend that the authors offer more detailed experimental evidence to substantiate these claims. Specifically, it would be valuable to include data demonstrating the sensor's real-world performance in detecting very low concentrations of gases or single molecules. Furthermore, comparisons with other established sensing methods or devices would be beneficial to validate the performance and highlight the advantages of the proposed sensor.

Version 1:

Reviewer comments:

Reviewer #1

(Remarks to the Author)

All my concerns have been well addressed. Therefore I support the publication of this manuscript.

Reviewer #2

(Remarks to the Author)

The authors have sufficiently addressed my comments. I recommend the manuscript for publication in its current form.

Reviewer #3

(Remarks to the Author)

The authors have addressed my concerns more than satisfactory, thus I recommend manuscript to be published in current form.

Reviewer #1

Main comment: This manuscript introduces a refractive index imaging sensor utilizing SPP structures of varying thicknesses. The differential guided-mode resonance, achieved through precise adjustment of structure thickness, results in high sensitivity to changes in the surrounding refractive index. Additionally, the authors showcase the potential application of this method in portable devices and propose an alternative low-cost fabrication process. This work is of significant interest to the research community focused on image sensors and SPP devices. Therefore, I support the publication of this manuscript, provided my concerns are addressed.

Reply: Thank you for your positive opinions and taking time to provide such valuable comments. Following is our one-by-one response.

Comment 1: My primary concern is regarding the manuscript's claim of "without sensitivity-range tradeoff." With a limited number of CCD pixels, there is always a tradeoff between sensitivity and the range of the proposed imaging sensor. While the measurement range of the sensor can be adjusted by the incident angle, as demonstrated in the manuscript, this constitutes a reconfiguration of the sensor rather than an elimination of the sensitivity-range tradeoff. The incident angle should be considered as part of the sensor's configuration. For instance, in a grating-based spectrometer with limited CCD pixels, finer spectral resolutions can be achieved using gratings with larger linear dispersions, but the CCD size limits the measured spectral range. Similarly, the measured spectral range can be extended by varying the incident angle, which is a reconfiguration of the spectrometer.

Reply and modifications: We appreciate your insightful comment regarding the sensitivity-range tradeoff in our proposed image sensor.

In our design, the measurement range can be dynamically adjusted by reconfiguring the incident angle, enabling flexible operation without increasing the total number of CCD pixels. Even though the measurement range can be extended with the fixed size of CCD, but as you pointed out, it's not suitable to describe it as "without" sensitivity-range tradeoff.

Per your suggestions, we have revised the manuscript to better articulate this point and avoid the impression that the sensitivity-range tradeoff is entirely eliminated. Instead, we will

emphasize that our system provides a flexible means to overcome or mitigate this tradeoff through reconfiguration, as demonstrated in the manuscript.

Corresponding, we also have changed the title of the manuscript to “*Ultrasensitive Imaging-based Sensor Unlocked by Differential Guided-Mode Resonance*”.

Comment 2: The schematic diagram (Figure 1) is somewhat confusing. This diagram depicts a structure with concentric rings, which aligns with the lithography-free method shown in Figure 4. However, Figure 2 depicts a pixelated structure, making the explanation of the device design rather unclear.

Reply and modifications: Thank you very much for pointing out this potential source of confusion.

The similar parts are: Figure 1 is conceptually aligned with Figure 4, while Figures 2 and 3 primarily emphasize the pixelated structure. As illustrated in Figure 1, the core principle of our approach is to achieve differential guided-mode resonance by regulating thickness differences. Although the designs in Figures 2 and 4 are presented in different forms, their fundamental concept remains the same—constructing thickness difference.

The different parts are: Figures 2 and 3 focus on discrete thickness differences, represented as a pixelated structure, whereas Figure 4 highlights a continuous thickness variation enabled by a lithography-free method. Thus, we start with the pixelated structure to demonstrate the concept of our approach and then to use the lithography-free method to construct the continuous structure with small thickness gradient.

We greatly appreciate your thoughtful feedback on this matter. To enhance clarity and prevent potential misunderstandings, we have added the following explanatory statement before Figure 2 in the revised manuscript:

“We start with the pixelated structure with thickness modulation to demonstrate the concept of differential guided-mode resonance for refractive index sensing”.

We have also added the following explanatory statement after Figure 4 in the revised manuscript (Line 250, Page 12):

“Note that Figure 4 highlights a continuous thickness variation enabled by a lithography-free method (aligned with conceptual Figure 1) while Figures 2 and 3 focus on discrete thickness

differences, represented by a pixelated structure. These four figures are presented in different forms, but their fundamental concept remains the same—constructing thickness difference.”

Comment 3: Additionally, please provide more details on the sensitivity of the imaging sensor to variations in the incident angle and the collimation of the incident waves.

Reply and modifications: Thank you for your valuable comment.

To address your concern, we have added the following detailed calculations about the sensitivity to variations in the incident angle and the collimation of the incident waves in the

Supporting Information (Section 17):

As shown in Figure 1b, the sensitivity S is defined as:

$$S = \frac{2|R_B - R_A|}{|n_B - n_A|} \quad (\text{S23})$$

It can also be expressed as:

$$S = \frac{2\Delta R}{\Delta n} \quad (\text{S24})$$

In Figure 2e, the relationship between the refractive index n and the thickness t at resonance is given as:

$$n = f(t) \quad (\text{S25})$$

Using Eq. S25, Eq. S24 can be reformulated as:

$$S = \frac{2\Delta R}{\Delta f(t)} = \frac{2\Delta R}{\Delta t} \frac{1}{f'(t)} = C \frac{1}{f'(t)} \quad (\text{S26})$$

Here, C is a constant determined by the fixed structure of the sensing chip. The sensitivity S is inversely proportional to $f'(t)$. Moreover, the influence of the incident angle is reflected in $f'(t)$.

In Figure 2g, the relationship curve is plotted under a fixed incident angle. Through further calculations, we obtained relationship curves of $1/f'(t)$ for different incident angles. We have also added a new figure (**Figure S28** in the **Supporting Information (Section 17)**) to show $1/f'(t)$ values (the sensitivity S is inversely proportional to $f'(t)$) remain nearly identical under different incident angles in the low refractive index region, while in the high refractive index region, sensitivity increases as the incident angle decreases.

Figure S28. The relationship between $1/f(t)$ and the refractive index under different incident angles.

The collimation of the incident wave would have some impact on the $1/f(t)$ value, as shown in **Figure S28**. However, the typical sensitivity is determined by the average of the maximum and minimum incident angles. In the low refractive index region, the non-collimation has minimal impact on sensitivity, whereas in the high refractive index region the average sensitivity would increase (even if the sensitivity difference would increase, as shown in the high refractive index region in **Figure S28**). Correspondingly, we have added the following text (Line 331, Page 15):

“Moreover, we also include a detailed discussion on the sensitivity of the imaging sensor to variations in the incident angle and the collimation of the incident waves (Supporting Information, Section 17).”

“The collimation of the incident wave would have some impact on the $1/f(t)$ value, as shown in Figure S28. However, the typical sensitivity is determined by the average of the maximum and minimum incident angles. In the low refractive index region, the non-collimation has minimal impact on sensitivity, whereas in the high refractive index region the average sensitivity would increase (even if the sensitivity difference would increase, as shown in the high refractive index region in Figure S28).” [(Supporting Information, Section 17)]

Regarding the sensitivity performance, to improve the sensitivity performance in previous Figure 5(ii), we have carried out a new experiment and adopted an optimized modification strategy and used streptavidin-biotin. The new result is added in **Figure 5**. The new experimental result shows a more evident pixel shift ~ 20 pixels, and the theoretical detection

limit can be estimated as small as 15 pM. We have added detailed modification methods, procedures, and results in the **Supporting Information (Section 15)**, as outlined below.

“15. The detailed modification and some results regarding the biotin test in Figure 5a

The surface modification process involved: Firstly, conducting silanization treatment to the deposited SiO₂ using 2% (v/v) APTES (3-aminopropyltriethoxysilane, C₉H₂₃NO₃Si) for 40 min. Secondly, conducting glutaraldehyde (5% (v/v)) treatment for 1 hour. Thirdly, conducting streptavidin incubation (200 µg/mL) for 1 hour. Then blocking other binding sites with 5 mg/mL BSA (bovine serum albumin) for 1 hour. Finally, monitoring the biotin solution (1 nmol/L) binding for 50 min.

We conducted real-time monitoring of the 1 nmol/L biotin binding process, with the resulting resonance stripe shift shown below (Figure S23). The stripe moves fast to the left at the beginning, that means the increasing refractive index of the surface. Then the stripe moves gradually as the time increasing.

Figure S23. Real-time monitoring of the 1 nmol/L biotin binding process. The red dashed line represents the movement of the stripes, while the white dashed line represents the background that does not move for reference.

As shown in Figure S24, the stripe curve of image gray shows a clear shift as the time increases during the binding process.

Figure S24. The stripe curve of image gray during the binding process. The arrow in the figure means the direction of time increasing, following with the peak shift.

The pixel shift curve for 1 nmol/L biotin is presented in Figure S25. A significant shift of over 20 pixels was observed at this low concentration. The binding signal increased rapidly at the initial stage and gradually reached saturation over time.

Figure S25. The pixel shift curve for 1 nmol/L biotin binding process.”

Reviewer #2

Main comment: In this manuscript, the authors introduced an innovative, imaging-based refractive index sensor employing differential guided-mode resonance (dGMR) approach to achieve ultra-high sensitivity without the typical tradeoff between sensitivity and dynamic range. By modulating the thickness of the waveguide layer at the nanometer scale, this sensor translates refractive index changes into highly sensitive imaging patterns, reaching sensitivities of up to million-level pixels per refractive index unit. This result represents a significant advancement over previous studies. I recommend that this manuscript be considered for publication following moderate revisions. Below, I have outlined several points for the authors to address.

Reply: Thank you for your positive opinions and taking time to provide very valuable comments. Our point-by-point response is below.

Comment 1: Metasurface-based refractive index sensors are a well-explored area of study. To provide a suitable benchmark, the authors should briefly compare image-based sensors and other types of refractive index sensors in the introduction, such as those based on spectrum shifting, spectrum splitting, phase changes, and efficiency changes. This comparison would help contextualize the performance and advantages of the proposed sensor within the broader field.

Reply and modifications: We appreciate your suggestion to include a comparison between image-based refractive index sensors and other types of refractive index sensors.

We have added a brief comparison in the introduction section of the manuscript. This comparison highlights the key differences in working principles, measurement mechanisms, and application scenarios between these sensor types. Specifically, we emphasize how image-based sensors offer unique advantages such as ease of data acquisition, and potential for compact, real-time sensing.

Correspondingly, we have added the following text in the introduction:

“Refractive index sensors based on spectral shifts or splitting require high-resolution spectrometers, making them bulky, expensive, and unsuitable for portable or real-time applications^{17,18}. Their sensitivity depends on the optical resonance structure, and the

resolution is limited by the spectrometer. Phase-based sensors offer high sensitivity but require complex setups and precise phase detection, limiting their portability and real-time usage^{19,20}. Intensity-based sensors monitor refractive index changes by detecting reflectance/transmittance variations. While simpler in design, they have lower sensitivity and are prone to noise and environmental interference. In contrast, an imaging-based sensor directly captures spatial light distributions or color changes, eliminating the need for complex instruments and making it more suitable for real-time and portable applications.”

We have also added a new table for the detailed comparison in the **Supporting Information (Section 18)**:

“18. The comparison between image-based refractive index sensors and other types of refractive index sensors

Table S1. The comparison between image-based sensors and other types of ones.

Type		Sensitivity	LOD	Preparation complexity	System complexity
Spectrum shifting/splitting^{9,10}		Low	High	Medium	Medium
Phase changes^{11,12}		High	Low	Medium	High
Amplitude changes^{13,14}		Medium	Medium	Medium	Low
Image-based sensors	Other types^{15,16}	Low (10³ pixel/RIU)	High	High	Low
	Our work	High (10⁶ pixel/RIU)	Low	Low	Low

Notes: LOD means limit of detection.”

Comment 2: The manuscript does not clearly explain the physical mechanism behind the remarkable sensitivity of “990,000 pixels/RIU, surpassing existing counterparts by nearly three orders of magnitude.” It is unclear if this sensitivity is primarily due to the unique differential guided-mode resonance (dGMR) mode. A more detailed comparison with existing image-based sensors, particularly references 34 and 35, would strengthen the paper.

Reply and modifications: We very much appreciate your insightful comments regarding the sensitivity mechanism and the comparison with existing image-based sensors.

For dGMR (differential guided-mode resonance). The key aspect is that it is differential representing the difference between the two resonances. As illustrated in Figure 1b, we construct two guided-mode resonances with slight spatial differences in thickness. These two structures form a sensing unit that enables the detection of refractive index changes, by mapping the resonance event to the spatial change of the stripes. As the refractive index changes, the resonance occurs from one guided-mode resonant structure to the other guided-mode resonant structure and one can detect the refractive index change through spatial shifts of resonance stripes.

Next, we elaborate on the physical mechanism behind the high sensitivity and provide a detailed comparison with existing image-based sensors, particularly references 34 (Adv. Mater. 33, 2100270 (2021).) and 35 (Small Methods 8, 2300873 (2024).). As shown in Figure 1b, we derive the sensitivity expression based on the dGMR sensing framework.

$$S = \frac{2|R_B - R_A|}{|n_B - n_A|} \quad (\text{S23})$$

It can also be expressed as:

$$S = \frac{2\Delta R}{\Delta n} \quad (\text{S24})$$

In Figure 2e, the relationship between the refractive index n and the thickness t at resonance is given as:

$$n = f(t) \quad (\text{S25})$$

Using Eq. S25, Eq. S24 can be reformulated as:

$$S = \frac{2\Delta R}{\Delta f(t)} = \frac{2\Delta R}{\Delta t} \frac{1}{f'(t)} = C \frac{1}{f'(t)} \quad (\text{S26})$$

Here, C is a constant determined by the fixed structure of the sensing chip. This sensitivity depends on two factors: the intrinsic sensing performance of the resonance structure and the spatial thickness difference (or gradient). As illustrated in Figure 4a, the smaller spatial thickness gradient significantly enhances the sensing sensitivity.

As shown by the theoretical framework in this work, the sensitivity in the gradient structure is inversely proportional to the gradient of the structures. However, previously published works lack a method to precisely control or build a low gradient profile. As a result, these approaches generally exhibit low sensitivity. The sensitivity of these sensors in the references 34 (Adv. Mater. 33, 2100270 (2021).) and 35 (Small Methods 8, 2300873 (2024).) is typically around

the 10^3 pixel/RIU level, exhibiting lower sensing performance due to their spatial geometric gradients, ultimately resulting in sensitivities that are three orders of magnitude worse than that of our proposed dGMR sensor.

Correspondingly, we have added the following text (**Supporting Information, Section 19**) is as follows:

“19. Detailed explanation of dGMR and the physical mechanism behind high sensitivity

dGMR refers to differential guided-mode resonance, where the critical component is the differential mechanism. As shown in the schematic diagram of Figure 1b, we construct two guided-mode resonances with slight spatial differences in thickness to form a sensing unit. This enables the detection of refractive index changes through spatial shifts of resonance fringes. Based on the dGMR sensing mechanism and theoretical framework, we derive the sensitivity expression. Sensitivity depends on the resonance structure's intrinsic sensing performance and the spatial thickness difference (or gradient). As shown in Figure 4a, the smaller spatial thickness gradient greatly enhances the sensing sensitivity.

Previous works^{34,35} involve the construction of geometric metasurfaces. Analyzing their systems under the dGMR framework reveals that their resonance structures have insufficient sensing performance, and their spatial geometric gradients are too large. As a result, their sensitivity is three orders of magnitude lower than that of the proposed dGMR sensor.”

Comment 3: Additionally, given that the detector’s pixel density may significantly impact this value, the authors should also discuss this factor to provide a more rigorous result.

Reply and modifications: Thank you for this insightful comment.

The detector used in our study is a standard CMOS camera with a pixel density of 1608×1104 pixels, as noted in the caption of Figure 4. This pixel density is typical for CMOS cameras and does not fall into the category of high-resolution CMOS cameras. As comparison, the pixel density in references 34 (Adv. Mater. 33, 2100270, 2021) and 35 (Small Methods 8, 2300873, 2024) is 5544×3694 . Therefore, the high sensitivity of our proposed sensor is not due to the use of a high-pixel-density camera but rather the inherent sensitivity of the sensor itself.

As the reviewer pointed out, pixel density does indeed influence sensitivity, and generally, sensitivity increases linearly with pixel density. However, higher pixel densities result in

smaller pixel sizes, which can reduce the signal intensity of the sensor (larger noise). Hence, higher pixel density is not always better. Additionally, we chose to use the unit pixel/RIU to describe sensitivity for two reasons: first, it allows for a straightforward comparison with other image-based sensors; second, pixel displacement is more intuitive and aligns better with the practical application scenarios of image-based sensors. To prevent potential misunderstandings, we have added the content that includes a conversion of sensitivity units in **Supporting Information (Section 11.1)**. This ensures that the reported value is independent of the camera's pixel density, making the results more robust and universally applicable:

“Additionally, to present the sensitivity independently of the detector's pixel density (CMOS camera with 1608×1104 pixels), we can convert the pixel-shift value to an absolute length-shift value. Given that each pixel corresponds to approximately $5 \mu\text{m}$, the sensitivity S ($990,000$ pixels/RIU) can be recalculated as $4,950,000 \mu\text{m}/\text{RIU}$.”

Comment 4: To expand the refractive index detection range, the authors introduced the incident angle as a new degree of freedom. However, if the structural dimensions were sufficiently large, for example, 8 inches instead of 4 inches as mentioned in the manuscript, could it be possible to cover a wide refractive index range without altering the incident angle? The authors should consider comparing these two approaches, especially regarding the potential errors and calibration issues associated with adjusting the incident angle. This comparison would provide a more comprehensive assessment of the method's feasibility and stability in practical applications.

Reply and modifications: Thank you for this valuable suggestion. It is absolutely correct that increasing the chip size, such as using an 8-inch chip, could broaden the refractive index detection range. This is because a larger size would result in a wider thickness distribution of guided-mode resonance. This relationship is also evident from Eq. S18 in the **Supporting Information (Section 11.2)**, as shown below:

$$\Delta n = \frac{L}{2C_n} \nabla t \quad (\text{S18})$$

Here, L is the chip size, indicating that the refractive index range is proportional to the chip size. We would also like to clarify that the 4-inch size mentioned in the manuscript refers to the wafer size used for batch fabrication, not the size of an individual chip. The dimensions of our

fabricated chip are 25.4 mm×25.4 mm.

As suggested by the reviewer, to provide a more comprehensive assessment of the method, we have added a discussion in the **Supporting Information (Section 11.2)** comparing the approach of increasing chip size with the method of adjusting the incident angle. The added content is as follows:

“From Eq. S18, we can see that increasing the chip size can also expand the refractive index detection range. However, in many applications the sensing area (i.e., the size of the sensing chip) is fixed or limited, we would prefer to tune the incident angle as a convenient way to extend the refractive index range, facilitating system miniaturization and integration. However, adjusting the incident angle introduces potential measurement errors, particularly during the transition to the next measurement cycle when the angle is reset to return the resonance fringes or rings to their initial position. To mitigate such errors, a possible correction method could involve marking the initial and final positions on the chip. When the camera detects that the resonance rings have reached the final position, an automated control algorithm could adaptively adjust the incident angle to the next measurement cycle, thereby reducing errors to some extent.”

Comment 5: In Figure 4, the authors employed the PECVD method, using deposition errors to introduce thickness difference in the thin film, which is an interesting approach. However, does this imply that the fabrication uncertainty may be higher compared to lithography? How did the authors control the fabrication parameters to achieve the desired thin-film thickness gradient? Additionally, what is the fabrication repeatability of the thin-film thickness gradient in the sensors? These factors are crucial for the accuracy and reproducibility of sensing performance.

Reply and modifications: Thank you for your insightful comments. We appreciate your recognition of our PECVD-based approach. In our method, the thickness variations (or thickness gradients) in the sensor chips are intentionally introduced by leveraging the intrinsic deposition characteristics of PECVD. We acknowledge that our previous version of the manuscript lacked a detailed discussion on fabrication repeatability. To address this, we have carried out additional experiments to verify the reproducibility of PECVD-based fabrication and included a more comprehensive analysis.

Specifically, we fabricated five sensor chips from the same batch under identical deposition conditions (gas flow rate, deposition power, and deposition time). The results confirm a high fabrication consistency, with a typical standard deviation of less than 2.4%. Correspondingly, in the revised manuscript, we have added new **Figures S29, S30 and S31** (on the reconstructed thickness profiles of five samples from the same batch), as well as the detailed additional experimental validation and discussion on experimental consistency in **Supporting Information (Section 20)**, which is as follows.

“20. The repeatability assessment for the sensor chip fabrication based on PECVD

To evaluate the repeatability of our PECVD-based fabrication process, we prepared five sensor chips from the same batch under identical deposition conditions, maintaining fixed gas flow rates, deposition power, chamber pressure, and a consistent deposition duration of 2 minutes and 30 seconds, as shown in Figure S29.

Figure S29. Five gradient sensor chips fabricated via PECVD from the same batch. From left to right, they are labeled as Sample A to Sample E. The scale bar represents 1 cm.

As demonstrated in Figure 4d and detailed in Supporting Information (Section 11.1), the sensitivity of the sensor chip, under identical incident angle and refractive index conditions, is primarily determined by the engineered thickness gradient. Thus, verifying the reproducibility of the PECVD deposition process directly translates to assessing the uniformity and consistency of the thickness gradient across multiple fabrication samples, which also means the repeatability of the thickness fabrication based on PECVD deposition.

To further confirm the repeatability of samples from the same batch, we reconstructed the thickness profile using angle-resolved measurements, following the approach described in Figures 3d and 4a. Five sensor chips were analyzed using angular scanning, and the corresponding imaging results at the same incident angle are shown in Figure S30. All five

chips exhibit identical resonance ring features, providing preliminary validation of the reproducibility of thickness-gradient sensors fabricated via PECVD.

Figure S30. The imaged resonance ring-stripe of five sensor chips from the same batch under the same incident angle. (a)-(e) is corresponding to the sample A-E. (f). The schematic diagram of the partial test system.

Figure S31 presents the reconstructed thickness profiles along a horizontal cross-section, revealing high consistency among the five samples. At the middle area (point P) of Figure S31, the fabrication standard deviation among the five chips is 2.4%, indicating a good reproducibility in our PECVD-based fabrication method.

Figure S31. The reconstructed thickness profiles of five samples from the same batch.”

We acknowledge that techniques such as photolithography and electron beam lithography (EBL) allow for more precise thickness control. Indeed, in Figure 3, we employed EBL to fabricate structures with controlled thickness variations. However, PECVD-based deposition offers distinct advantages, including ease of fabrication, high throughput, and scalability. Given that PECVD inherently exhibits minimal deposition errors, it provides a practical route for producing sensor chips with subtle thickness differences. One can refine control over deposition parameters—such as gas flow rate, deposition power, and substrate morphology—to achieve even finer thickness gradients and further enhance sensor performance.

Comment 6: Recent published papers about resonance engineering or meta-sensing can be considered to be involved in References, e.g., Nature Communications volume 15, 9658 (2024); ACS Nano, 11598-11618 (2022).

Reply and modifications: Thank you for your valuable suggestion. We have added these relevant references in the introduction (Paragraph 1, Refs. 6 and 7).

Comment 7: In Figure 5a(ii), the pixel shift after each step appears unclear. Shouldn't the pixel shift be cumulative, reflecting the overall change in refractive index across the sensor?

Reply and modifications: Thank you for pointing this out.

In Figure 5a, different solutions are used for each modification/measurement step, leading to significant differences in refractive indices, i.e., notable variations in the background refractive index. For example, Tris-buffer is used in the first modification step, while PBS buffer is used in the second step. Therefore, we chose to measure the pixel shift (Δ pixel) in real time from the moment the solution was introduced, reflecting the relative pixel shift compared to the start of each step. We have added the following clarification in the **Supporting Information (Section 15)**: “*The pixel shift means the relative shift pixels compared to the initial process.*”.

Comment 8: Additionally, the curve fitting for R-IgG seems less accurate compared to the other three graphs. The authors should provide a brief analysis of this discrepancy to clarify the potential causes and implications for the sensor's performance.

Reply and modifications: Thank you for your insightful comment.

The previous surface modification approach may have had low efficiency, which has led to

a less pixel shift for R-IgG process. To improve binding performance, in the revised manuscript, we adopted an optimized modification strategy and used streptavidin-biotin, known for its strong affinity, as the detection target (Biosens. Bioelectron. 2015, 67: 230-236; Phys. Chem. B 2009, 113, 25, 8776-8783; Anal. Chem. 2010, 82, 12, 5211-5218). The new result is added to Figure 5. The new experimental result shows a more evident pixel shift ~ 20 pixels, and the theoretical detection limit can be estimated as small as 15 pM.

We conducted real-time monitoring of the 1 nmol/L biotin binding process, with the resulting resonance stripe shift shown in new **Figure S23**. The stripe shifts fast to left at the beginning, and moves gradually as the time increasing, means the increasing refractive index of the surface.

Figure S23. Real-time monitoring of the 1 nmol/L biotin binding process. The red dashed line represents the movement of the stripes, while the white dashed line represents the background that does not move for reference.

As shown in new **Figure S24**, the stripe curve of image gray shows a clear shift as the time increases during the binding process.

Figure S24. The stripe curve of image gray during the binding process. The arrow in the figure means the direction of time increasing, following with the peak shift.

The pixel shift curve for 1 nmol/L biotin is presented in the newly added **Figure S25**. A significant shift of over 20 pixels was observed at this low concentration. The binding signal increased rapidly at the initial stage and gradually reached saturation over time.

Figure S25. The pixel shift curve for 1 nmol/L biotin binding process.

Based on the calculated pixel shift standard deviation (~ 0.3 pixels), which mentioned in the **Supporting Information (Section 11.3)**, the detection limit can be estimated as small as 15 pM ($1 \text{ nM}/(20/0.3)$).

We have **revised Figure 5** and added a corresponding description (Page 14, above Figure 5) as follows:

“

Figure 5. The surface sensing and monitoring and the portable device designed for this thickness-modulated refractive index sensor. **(a)** The monitoring response curve of the biotin molecule binding event (the concentration is 1 nmol/L). The insert is the surface modification process. APTES means 3-aminopropyltriethoxysilane. GA means glutaraldehyde. SA means streptavidin. BSA means bovine serum albumin. **(b)** The portable device design and its application. (i) A portable prototype ($20 \times 14 \times 8 \text{ cm}^3$) based on a plug-and-play sensor chip. (ii) Different ring-stripe images under various solution conditions. **(c)** (i) Two-dimension humidity sensing based on this thickness-modulated chip. (ii) The ring-stripe images as the humidity increases in a chamber, captured by the smartphone camera.”

“The high-specificity binding between streptavidin and biotin is a well-established model in biomolecular recognition, playing a crucial role in medical diagnostics, food safety, and environmental monitoring. As shown in Figure 5a, we functionalized the sensing surface by silanization and streptavidin incubation to enable biotin detection. The detailed modification steps are provided in the Supporting Information (Section 15). Even at a low biotin concentration of 1 nmol/L, our sensor exhibits a signal response exceeding 20 pixels.”

Additionally, we have included detailed modification methods, procedures, and results in the

new added **Supporting Information (Section 15)**, as outlined below.

“15. The detailed modification and results regarding the biotin test in Figure 5a

The surface modification process involved: Firstly, conducting silanization treatment to the deposited SiO₂ using 2% (v/v) APTES (3-aminopropyltriethoxysilane, C₉H₂₃NO₃Si) for 40 min. Secondly, conducting glutaraldehyde (5% (v/v)) treatment for 1 hour. Thirdly, conducting streptavidin incubation (200 µg/mL) for 1 hour. Then blocking other binding sites with 5 mg/mL BSA (bovine serum albumin) for 1 hour. Finally, monitoring the biotin solution (1 nmol/L) binding for 50 min.

We conducted real-time monitoring of the 1 nmol/L biotin binding process, with the resulting resonance stripe shift shown below (Figure S23). The stripe moves fast to left from the beginning, that means the increasing refractive index of the surface. Then the stripe moves gradually as the time is increasing.

Figure S23. Real-time monitoring of the 1 nmol/L biotin binding process. The red dashed line represents the movement of the stripes, while the white dashed line represents the background that does not move for reference.

As shown in Figure S24, the stripe curve of image gray shows a clear shift as the time increases during the binding process.

Figure S24. The stripe curve of image gray during the binding process. The arrow in the figure means the direction of time increasing, following with the peak shift.

The pixel shift curve for 1 nmol/L biotin is presented in the Figure S25. A significant shift of over 20 pixels was observed at this low concentration. The binding signal increased rapidly at the initial stage and gradually reached saturation over time.

Figure S25. The pixel shift curve for 1 nmol/L biotin binding process.”

Reviewer #3

Main comment: The authors present an ultrasensitive imaging-based sensor with large sensitivity range using guided-mode resonance, which seems to be promising in the field of imaging-based sensors. This manuscript could be published if the authors can clarify following questions.

Reply: Thank you for your positive opinions and taking time to provide very valuable comments. Our point-by-point response is below.

Comment 1: Studies have reported on target detection methods utilizing imaging-based surface plasmon resonance (SPR) (e.g., *Analytical Chemistry*, 2001, 73(22): 5525-5531; *Lab on a Chip*, 2007, 7(9): 1206-1208). Additionally, there is a body of work addressing the achievement of ultrasensitive sensing through the modulation of the geometric morphology of plasmonic sensors (e.g., *Nanoscale*, 2019, 11, 12471). Given the existing literature, could you elaborate on the unique innovation in your approach, particularly with respect to the regulation of height changes in the geometric morphology of the sensor?

Reply and modifications: Thank you for highlighting these studies and raising this important question. We appreciate the opportunity to clarify the unique innovation of our approach.

Regarding the literature you mentioned, the first two papers discuss imaging-based sensors using simple intensity/amplitude-based SPR refractive index sensors. The sensitivity of these sensors is determined by the resonance mode (the mismatch between the SPP propagation wave vector and incident wave vector) and its sensitivity to changes in the external refractive index. However, their sensitivity is limited by the inherent properties of the resonance mode. The third paper is similar to the gradient-type sensors ([*Adv. Mater.* 33, 2100270 (2021)], [*Small Methods* 8, 2300873 (2024)]) we mentioned in the introduction, as they all rely on constructing gradient structures (e.g., plasmonic gradient structures) for spatial mapping. As shown by the established theoretical framework in the present work, the sensitivity in the gradient structure is inversely proportional to the gradient of the structures. However, in the previous cases, there is no method to precisely control or build a low gradient profile. As a result, these approaches generally exhibit lower sensitivity and *the sensitivity of these sensors is typically around the 10^3 pixel/RIU level*. Furthermore, a comparison shows that while the sensitivity of their sensor is still

significantly lower by at least two orders of magnitude than the sensitivity of our proposed sensor.

Moreover, the regulation of thickness in our design is not just a geometric optimization but an integral part of the sensing mechanism. As demonstrated in our results, the differential thickness between neighboring regions creates a spatially varying guided-mode resonance, enabling highly sensitive and dynamic refractive index detection.

To provide additional clarity, we have included a simple comparison of our approach with the referenced methods in the introduction part of the revised manuscript.

“Some imaging-based SPR sensors^{40,41} combine imaging tools for refractive index sensing. These sensors are essentially simple intensity-based SPR sensors, where the sensitivity is limited by the inherent sensitivity of the resonance mode. Recent studies have explored imaging-based refractive index sensors through the construction of geometric metasurfaces⁴²⁻⁴⁷. These studies use gradient structures (e.g., plasmonic gradient structures^{42,43,48}) for spatial mapping. However, there lack of a method to precisely control or build a small gradient profile. As a result, these approaches generally exhibit low sensitivity. The sensitivity of these sensors is usually around 10^3 pixel/RIU level^{42,43}.”

Comment 2: From the results presented, your sensor appears to demonstrate an ultra-high sensitivity of 990,000 pixels/RIU. However, upon a thorough review of the manuscript, this value seems to correspond more closely to a theoretical calculation of sensitivity. Typically, refractive index sensitivity is calculated using the formula ΔP (pixel shift) / Δn (change in refractive index).

Reply and modifications: Thank you for your observation and for highlighting this important aspect. We apologize for any confusion we may have caused in the previous version. The sensitivity of 990,000 pixels/RIU presented in this paper is an experimental value, not a theoretical calculation. As shown in Figure 4f, this sensitivity is derived using the method you mentioned: refractive index sensitivity = ΔP (pixel shift) / Δn (change in refractive index), which, in this case, is 743 pixels / 0.00075 RIU, resulting in approximately 990,000 pixels/RIU. For a theoretical estimate, based on the curve trend, the sensitivity would actually be slightly higher than 990,000 pixels/RIU. To avoid further misunderstanding, we have revised the main

text to clarify that this value is experimentally obtained, not theoretically calculated (Line 267, Page 12):

“The maximum experimental sensitivity can reach 990000 pixel/RIU.”

Comment 3: Could you clarify why the ratio between pixel shift and refractive index in your S9 formula exhibits a two-fold relationship?

Reply and modifications: Thank you for your question.

The two-fold relationship between pixel shift and refractive index in formula (S9) is a result of that we measure the diameter of the ring rather than its radius. Since the diameter is twice the radius, this naturally results in a factor of two. This also can be seen from formula (S9):

$$S = \frac{2(P_2 - P_1)}{n_2 - n_1} \quad (S9)$$

Here, P_1 and P_2 are the position of the ring stripe pattern (i.e. the radius of the ring stripe) under different refractive index conditions (n_1 and n_2), respectively. However, since we measure the pixel shift based on the ring's diameter rather than its radius, this introduces a two-fold relationship.

We appreciate your insightful question and hope this explanation clarifies the reasoning behind this factor. To avoid misunderstanding, we have added the following explanation in the **Supporting Information (Section 11.1)**:

“Since we measure the pixel shift based on the ring's diameter rather than its radius, this introduces a two-fold relationship in the formula above.”

Comment 4: Furthermore, the inclusion of height change in the sensitivity calculation is intriguing. Could you provide a more detailed rationale for incorporating this factor into the formula?

Reply and modifications: Thank you for your comment and for raising this important point.

The inclusion of height in the sensitivity calculation is based on our constructed dGMR structure. For dGMR (differential guided-mode resonance). The key aspect is that it is differential representing the difference between the two resonances. As illustrated in Figure 1b, we construct two guided-mode resonances with slight spatial differences in thickness, which means the height change. These two structures form a sensing unit that enables the detection of

refractive index changes, by mapping the resonance event to the spatial change of the stripes. As the refractive index changes, the resonance occurs from one guided-mode resonant structure to the other guided-mode resonant structure and one can detect the refractive index change through spatial shifts of resonance stripes. This is why we need to incorporate the height/thickness parameters related to spatial stripe variation into the sensitivity calculation formula.

Correspondingly, we have also added some content to give the following detailed explanation for the height change incorporation in **Supporting Information (Section 11.1)**:

“Furthermore, dGMR refers to differential of the guided-mode resonance, where the critical component is the differential nature, representing the difference between the two resonances. In this model, we use the height/thickness differences to create the dGMR mode (introducing the “d” in dGMR), which allows us to map changes in refractive index (i.e., the resonance condition of the GMR) to spatial changes in the resonance stripes. Different stripes have different heights/thicknesses of the GMR mode. This mapping reveals a novel mechanism for sensitivity enhancement, which is why we need to incorporate the height/thickness parameters related to spatial stripe variation into the sensitivity calculation formula.”

We have also added further details in the **Supporting Information (Section 19)** to provide more clarity on this, which is as follows:

“19. Detailed explanation of dGMR and the physical mechanism behind high sensitivity

dGMR refers to differential guided-mode resonance, where the critical component is the differential mechanism. As shown in the schematic diagram of Figure 1b, we construct two guided-mode resonances with slight spatial differences in thickness to form a sensing unit. This enables the detection of refractive index changes through spatial shifts of resonance fringes. Based on the dGMR sensing mechanism and theoretical framework, we derive the sensitivity expression. Sensitivity depends on the resonance structure’s intrinsic sensing performance and the spatial thickness difference (or gradient). As shown in Figure 4a, the smaller spatial thickness gradient greatly enhances the sensing sensitivity.”

Comment 5: When comparing your work with that in the literature on direct imaging of sensors based on changes in geometric morphology (e.g., *Advanced Materials*, 2021, 33(29): 2100270),

it appears that the sensitivity of the sensor in that study is approximately 500,000 pixels/RIU, based on the method you have applied. Could you please explain how your sensor demonstrates superior performance in comparison to this work? Specifically, what are the key advantages that make your sensor more sensitive?

Reply and modifications: Thank you for your valuable suggestion. We appreciate the opportunity to clarify how our sensor compares to the one discussed in the cited work. We would like to clarify that the sensitivity reported in the study (Advanced Materials, 2021, 33(29): 2100270) you mentioned is not 500,000 pixel/RIU but rather 1,040 pixel/RIU. To quote the original paper: “*Figure 5b indicates that the transmission dip moves towards the sensor center when the covering medium has a higher refractive index. By fitting the position of the transmission dips, the sensitivity S_m is estimated to be 1,040 pixel/RIU.*”. That paper used the camera with much higher pixel density than us (To quote the original paper: “*The exposure time of each image, which was composed of 5544×3694 effective pixels in 16 bit, was 5 ms (gain=30).*”). In contrast, our sensor achieves an experimental sensitivity of 990,000 pixel/RIU. This value represents a sensitivity nearly three orders of magnitude greater than that reported in the mentioned study. If the same pixel-density camera (as mentioned above in the reference) is used, its sensitivity would be even higher. Correspondingly, we have also added the following direct comparison in the main text (Line 267, Page 12):

“In the extended dynamic range, the maximum experimentally measured sensitivity of our sensor can reach 990000 pixel/RIU, as shown in Figure 4f, setting a record high sensitivity exceeding counterparts by nearly three orders of magnitude.”.

To give some explanations for this record-high sensitivity. We provide a detailed explanation of dGMR (differential guided-mode resonance). The key aspect of dGMR is that it is differential. As illustrated in Figure 1b, we construct two guided-mode resonances with slight spatial differences in thickness. These two resonances form a sensing unit that enables the detection of refractive index changes through spatial shifts of resonance fringes. As long as one can separate the two resonance structures as much as possible, that is, make the gradient as small as possible, one can get a larger sensitivity.

Next, we elaborate on the physical mechanism behind the high sensitivity and provide a detailed comparison with the sensors based on changes in geometric morphology (e.g.,

Advanced Materials, 2021, 33(29): 2100270). As shown in Figure 1b, we derive the sensitivity expression based on the dGMR sensing framework.

$$S = \frac{2|R_B - R_A|}{|n_B - n_A|} \quad (\text{S23})$$

It can also be expressed as:

$$S = \frac{2\Delta R}{\Delta n} \quad (\text{S24})$$

In Figure 2e, the relationship between the refractive index n and the thickness t at resonance is given as:

$$n = f(t) \quad (\text{S25})$$

Using Eq. S25, Eq. S24 can be reformulated as:

$$S = \frac{2\Delta R}{\Delta f(t)} = \frac{2\Delta R}{\Delta t} \frac{1}{f'(t)} = C \frac{1}{f'(t)} \quad (\text{S26})$$

Here, C is a constant determined by the fixed structure of the sensing chip. This sensitivity depends on two factors: the intrinsic sensing performance of the resonance structure and the spatial thickness difference (or gradient). As illustrated in Figure 4a, the small spatial thickness gradient significantly enhances the sensing sensitivity. In contrast, references based on changes in geometric morphology (e.g., Advanced Materials, 2021, 33(29): 2100270) lack an approach to precisely control the gradient to get a small gradient, ultimately resulting in sensitivities that are three orders of magnitude lower than that of our proposed dGMR sensor.

The newly added content (**Supporting Information, Section 19**) is as follows:

“19. Detailed explanation of dGMR and the physical mechanism behind high sensitivity.

dGMR refers to differential guided-mode resonance, where the critical component is the differential mechanism. As shown in the schematic diagram of Figure 1b, we construct two guided-mode resonances with slight spatial differences in thickness to form a sensing unit. This enables the detection of refractive index changes through spatial shifts of resonance fringes. Based on the dGMR sensing mechanism and theoretical framework, we derive the sensitivity expression. Sensitivity depends on the resonance structure’s intrinsic sensing performance and the spatial thickness difference (or gradient). As shown in Figure 4a, the smaller spatial thickness gradient greatly enhances the sensing sensitivity.

Previous works involve the construction of geometric metasurfaces. Analyzing their systems under the dGMR framework reveals that their resonance structures have insufficient sensing

performance, and their spatial geometric gradients are too large. As a result, their sensitivity is three orders of magnitude lower than that of the proposed dGMR sensor.”

We hope this addresses your concerns and strengthens the manuscript. Thank you again for your valuable comment and suggestion.

Comment 6: The manuscript does not provide refractive index sensitivity values for the test solutions presented in Figure 5. Since the resonance angle is expected to vary with the refractive index gradient, the nanoarray presented, which undergoes significant height changes, does not appear to exhibit continuous variation (as shown in Supporting Information S8, S9). Could you provide 2-3 SEM images at a magnification of $\times 5000$ to $\times 10000$, as well as AFM images at tilted angles, to characterize the uniformity and height variations of the sensor's microstructure? This would enable a more accurate calculation of the actual sensitivity based on data from the fabricated structures.

Reply and modifications: Thank you for your valuable feedback.

Firstly, we want to clarify that regarding the humidity response in Figure 5, our intention was to showcase an early-stage application of the portable sensing system, captured directly using a smartphone for real-world adaptability. Given the simplicity of our humidity chamber setup, precise humidity control was not feasible, and the data points in Figures 5(iii) and 5(iv) were randomly selected for illustrative purposes. However, despite these limitations, Figure 5 clearly shows a monotonic shift in the resonance fringes with humidity variation, confirming a consistent response rather than random fluctuations.

Secondly, we would like to clarify that in Figures 4 and 5, we constructed a silica thin-film structure with a continuously varying thickness. Regarding the Supporting Information Figures S8 and S9, it corresponds to the pixelated sensor structure in Figure 3. Therefore, we did not perform AFM or SEM characterization of the silica thin-film structure. However, we also conducted thickness reconstruction of the deposited silica film, obtaining its thickness profile, as shown in Figure 4a. The reconstruction method is detailed and validated in the Supporting Information (Sections 4 and 8), which mainly depends on the scanning resonance angle. These results confirm that the silica thickness varies continuously in Figure 4a.

Regarding sensitivity calculations, they were not derived theoretically but measured based

on fringe shifts caused by actual refractive index changes.

As suggested, in this revised manuscript, we characterized the exposed and developed SU-8 surfaces using AFM. As shown in Figure R1, the surface appears relatively smooth with minimal roughness.

Figure R1. The AFM images of the fabricated SU-8 resist square based on EBL patterning. (a) 2D AFM image. (b) 3D AFM image.

We have also done some additional experiments and provided experimental results on SU-8 dose-controlled thickness precision, as requested by this reviewer, based on the EBL patterning, showing that this method can control thickness variations within 3 nm (see thickness variations in the configuration of Figure 2a). This confirms its feasibility for constructing our thickness-gradient sensor. We fabricated SU-8 samples at lower exposure doses of 2.0, 2.5, 3.0, and 3.5 $\mu\text{C}/\text{cm}^2$, with four samples per dose to evaluate thickness variations. As shown in new **Figure S34** of the revised manuscript, we measured the thicknesses of these patches by AFM. The SU-8 thickness increases rapidly with increasing dose and then gradually slows down the increase as the dose is large enough. The thickness deviation among samples with the same dose is less than 3 nm, indicating that precise thickness control can be achieved by adjusting the exposure dose, enabling the fabrication of subtle thickness variations and gradients.

Figure S34. The relationship curve between the SU-8 exposure thickness and the exposure dose.

Correspondingly, the added content on the new **Figure S34** and additional experimental results on thickness-controlling precision is as follows (**Supporting Information, Section 22**):

“22. The thickness controlling experiment through EBL dose-modulation patterning

We fabricated SU-8 samples at lower exposure doses of 2.0, 2.5, 3.0, and 3.5 $\mu\text{C}/\text{cm}^2$, with four samples per dose to evaluate thickness variations. As shown in Figure S34, the SU-8 thickness initially increases rapidly with dose and then gradually slows down the increase as the dose continues to rise. The thickness deviation among samples with the same dose is less than 3 nm, indicating that precise thickness control can be achieved by adjusting the exposure dose, enabling the fabrication of subtle thickness variations and gradients.

Figure S34. The relationship curve between the SU-8 films thickness and the exposure dose.”

We appreciate your suggestions again and hope our reply can address your concerns.

Comment 7: The plot in Figure 4(f), which illustrates the relationship between pixel position and refractive index, presents only three data points. This limited data set is insufficient to provide convincing evidence of a reliable relationship. Additionally, it is unclear why two line segments with different slopes are used; could you clarify why higher data points were selected for the refractive index calculation? It would be more robust to test at least five data points to improve the reliability of the conclusion.

Reply and modifications: Thank you for your feedback and for raising this important point. We would like to clarify the following aspects:

To avoid any confusion, in the revised manuscript we have updated Figure 4(f) with eight data points to present a more complete data set, as shown below.

“

Figure 4. Ultrasensitive refractive index imaging sensor based on the ring stripe shift of the thickness-modulation chip through the dGMR. **(a)** The structure of the sensing chip and the measured and calculated thickness profiles. In the three-dimensional coordinate system (x - y - t), a point represents the thickness of the dielectric layer at position (x - y), and the spatial distribution of the thickness is characterized by the surface fitted by the discrete points. The image in the x - y plane is a contour map of the fitted thickness spatial distribution. **(b)** The refractive index sensing results. **(i)** The measured distinct ring pattern images under different concentrations. **(ii)** The curve between the pixel shift and concentration, as well as the sensitivity curve. **(c)** The sensing resolution analysis. **(d)** The analysis of sensitivity and dynamic range as thickness gradient varies. **(e)** The new measurable cycle, which can

widen the dynamic range of concentrations, from range (0, 1.0%) to additional range (1%, 2%), and so on, by using different incident angles in different measurement cycles. Dynamic range cycles can be continuously increased by simply re-adjusting the incident angle. **(f)** In the new dynamic range, the maximum experimental sensitivity can reach 990000 pixel/RIU (The resolution of the CMOS camera is 1608×1104 pixels). **(g)** The multi-channel sensing chip setup and the high-throughput imaging sensing with different solutions.”

Regarding the use of two-lines segments with different slopes in Figure 4(f), we would like to clarify that the purpose of these data points is to calculate the sensitivity, not to establish a calibration curve. The range of data points shown in original Figure 4(f) in the previous submission corresponds to the high refractive index range (1.0% concentration to 2.0% concentration). In this high refractive index range, we did not measure multiple data points at higher concentrations. As mentioned in your previous comment, the sensitivity is derived using the method you mentioned: refractive index sensitivity = ΔP (pixel shift) / Δn (change in refractive index), which, in this case, is 743 pixels / 0.00075 RIU, resulting in approximately 990,000 pixels/RIU.

Since we already obtained all the experimental parameters (ΔP and Δn) for the sensitivity calculation, the limited number of data points does not affect the accuracy of our sensitivity analysis. The different slopes in Figure 4(f) are used to illustrate the trend of pixel shift with respect to refractive index change, not to represent a precise calibration curve. Higher data points were selected for the sensitivity calculation because they correspond to the maximum sensitivity (experimental value), better showcasing the sensor’s performance.

Moreover, in calculating the sensitivity in Figure 4(f), we did not rely on fitting analysis or estimated data from a fitting curve. Instead, we used the experimental data directly, which results in a lower sensitivity estimate (since a fitting curve would likely produce a steeper slope).

Correspondingly, we have also added a note that “*The purpose of these data points is to calculate the sensitivity in the high refractive index range, not to establish a calibration curve.*” (Line 270, Page 13)

Comment 8: Regarding the pixel shift calculation shown in Figure 4(b), could you explain the method used to determine the pixel shift? Was this calculated through an averaging process

over the entire circle, or was a specific region analyzed?

Reply and modifications: Thank you for raising this insightful question.

We measured the pixel shift based on the diameter of the ring stripe in the horizontal direction (x-direction in **Figure S22**), as indicated by the red arrow, rather than averaging over the entire ring (in prism-coupled imaging, image compression occurs along the direction parallel to light propagation (y-axis in the figure)). To determine the pixel shift, we first performed grayscale analysis on the image to obtain the grayscale distribution curve along the x-direction. We then identified the pixel position corresponding to the lowest grayscale value after the moving average filter processing, which allowed us to accurately calculate the pixel shift.

Accordingly, we have added the corresponding content in the **Supporting Information (Section 14)** to explain **Figure S22** and improve the manuscript, which is as follows:

“14. The method to determine the pixel shift of the ring stripe.

We measured the pixel shift based on the diameter of the ring stripe in the horizontal direction (x-direction in the Figure S22), as indicated by the red arrow, rather than averaging over the entire ring. This choice was made because, in prism-coupled imaging, image compression occurs along the direction parallel to light propagation (y-axis in the figure), leading to an elliptical appearance of the ring. Thus, the diameter of the ring in the horizontal direction can reflect the real pixel shift.

To determine the pixel shift, we first performed grayscale analysis on the image to obtain the grayscale distribution along the x-direction. We then identified the pixel position corresponding to the lowest grayscale value as the starting or ending point, which allowed us to accurately calculate the pixel shift.”

Figure S22. The schematic diagram of pixel shift determination.

Comment 9: Moreover, based on the physical demonstration in Figure 4(g), it seems that the

fabricated template is not a perfectly regular circle, with observable variations in the thickness and placement of the stripes across different regions. Does this method exhibit reproducibility? To strengthen the analysis, could you provide error bars for the data to support the claims and clarify any potential variations?

Reply and modifications: Thank you for your valuable comments.

Firstly, we would like to clarify that Figure 4(g) represents the sensor chip integrated with a microfluidic system. The segmented pattern arises from the bonding of the sensor chip with a custom-designed fan-shaped microfluidic channel, which enables spatial partitioning by introducing different solutions into distinct regions. The observed stripe variations are a result of these different solutions. However, this figure primarily serves to demonstrate the compatibility of our sensor with microfluidic integration, rather than providing in-depth quantitative analysis. The detailed data analysis is presented in Figure 4(b).

Regarding the reproducibility of PECVD-based fabrication, we acknowledge the importance of verifying its consistency. Our approach leverages PECVD's intrinsic deposition variations to create controlled thickness gradients for sensing applications. To rigorously assess the repeatability, we have done some additional experiments and fabricated five sensor chips under identical deposition conditions (gas flow rate, power, and time), followed by thickness profile reconstruction. The results confirm a high fabrication consistency, with a typical standard deviation of less than 2.4%. Correspondingly, in the revised manuscript, we have added new Figures S29, S30 and S31 (on the reconstructed thickness profiles of five samples from the same batch), as well as the detailed additional experimental validation and discussion on experimental consistency in Supporting Information (Section 20), which is as follows

“20. The repeatability assessment for the sensor chip fabrication based on PECVD.

To evaluate the repeatability of our PECVD-based fabrication process, we prepared five sensor chips from the same batch under identical deposition conditions, maintaining fixed gas flow rates, deposition power, chamber pressure, and a consistent deposition duration of 2 minutes and 30 seconds, as shown in Figure S29.

Figure S29. Five gradient sensor chips fabricated via PECVD from the same batch. From left to right, they are labeled as Sample A to Sample E. The scale bar represents 1 cm.

As demonstrated in Figure 4d and detailed in Supporting Information (Section 11.1), the sensitivity of the sensor chip, under identical incident angle and refractive index conditions, is primarily determined by the engineered thickness gradient. Thus, verifying the reproducibility of the PECVD deposition process directly translates to assessing the uniformity and consistency of the thickness gradient across multiple fabrication samples, which also means the repeatability of the thickness fabrication based on PECVD deposition.

To further confirm the repeatability of samples from the same batch, we reconstructed the thickness profile using angle-resolved measurements, following the approach described in Figures 3d and 4a. Five sensor chips were analyzed using angular scanning, and the corresponding imaging results at the same incident angle are shown in Figure S30. All five chips exhibit identical resonance ring features, providing preliminary validation of the reproducibility of thickness-gradient sensors fabricated via PECVD.

Figure S30. The imaged resonance ring-stripe of five sensor chips from the same batch under the same incident angle. (a)-(e) is corresponding to the sample A-E. (f). The schematic diagram of the partial test system.

Furthermore, Figure S31 presents the reconstructed thickness profiles along a horizontal cross-section, revealing high consistency among the five samples. At point P of Figure S31, an error analysis was conducted, showing that under identical PECVD deposition parameters, the fabrication standard deviation among the five chips is merely 2.4%, indicating a good reproducibility in our PECVD-based fabrication method.”

Figure S31. The reconstructed thickness profiles of five samples from the same batch.”

Comment 10: This work achieves ultrasensitivity by fabricating thickness-tunable guiding resonators that align with the SPR resonance angle. However, the reported ultra-sensitivity appears to include variations in distance due to processing errors (e.g., deposition). To allow readers to more accurately assess the sensing performance, it would be helpful if the authors provided the height interval spacing, including the effects of fabrication errors. A reasonable spacing range would better inform the evaluation of the sensor's sensitivity.

Reply and modifications: Thank you for your valuable suggestion.

As shown in Figure 4d, the sensitivity is inverse proportional to the gradient of thickness (∇t), that is the height spacing.

For our sensor chips, we leveraged the inherent deposition variations in PECVD to create a

subtle thickness gradient. However, the thickness gradient itself is also influenced by radial position—smaller near the center and larger at the edges—which explains the observed spatial dependence of sensitivity, which can also be seen in new added **Figure S31**. Additionally, since PECVD produces a continuously varying silica film, Figure 4 does not involve discrete height intervals; the thickness changes smoothly across the surface.

Figure S31. The reconstructed thickness profiles of five samples from the same batch.

To further clarify the impact of fabrication variations, we have added a detailed discussion in **Supporting Information (Section 20)**, as mentioned before in our response to your previous comment.

“To further confirm the repeatability of samples from the same batch, we reconstructed the thickness profile using angle-resolved measurements, following the approach described in Figures 3d and 4a. Five sensor chips were analyzed using angular scanning, and the corresponding imaging results at the same incident angle are shown in Figure S30. All five chips exhibit identical resonance ring features, providing preliminary validation of the reproducibility of thickness-gradient sensors fabricated via PECVD.”

Figure S30. The imaged resonance ring-stripe of five sensor chips from the same batch under the same incident angle. (a)-(e) is corresponding to the sample A-E. (f). The schematic diagram of the partial test system.

Furthermore, Figure S31 presents the reconstructed thickness profiles along a horizontal cross-section, revealing high consistency among the five samples. At point P of Figure S31, an error analysis was conducted, showing that under identical PECVD deposition parameters, the fabrication standard deviation among the five chips is merely 2.4%, indicating a good reproducibility in our PECVD-based fabrication method.

Figure S31. The reconstructed thickness profiles of five samples from the same batch.”

Moreover, our approach is broadly applicable, as fabrication systems inherently exhibit

processing variations. One can refine control over deposition parameters, such as gas flow rate, deposition power, and substrate morphology, to achieve even finer thickness gradients and further enhance sensor performance.

Thank you again for your insightful feedback and hope this clarification addresses the concerns.

Comment 11: To the best of our knowledge, achieving precise control over the height of structures during nanofabrication remains a significant challenge. The manuscript does not clearly demonstrate how height-controllable fabrication was achieved, nor does it provide data or microstructural characterization to support the reliability and reproducibility of the proposed method. From Figures 5(iii) and (iv), the observed changes in the image with humidity appear irregular. In light of this, it raises concerns about the reliability of the sensitivity calculations based on these image changes.

Reply and modifications: Thank you for your valuable suggestion.

As you pointed out, precisely controlling structure height in nanofabrication is indeed a challenge. In Figure 3, when patterning discrete thickness variations using EBL, we employed a random dose modulation approach, where the final thickness of SU-8 resist after development correlates with the exposure dose. This dose-thickness relationship has also been explored in our previous work with different methods (Optics Express, 2019, 27(15): 21646-21651). Additionally, we have added **Supporting Information (Section 22)** with experimental validation of dose-controlled thickness tuning. In the experiment, we fabricated SU-8 samples at lower exposure doses of 2.0, 2.5, 3.0, and 3.5 $\mu\text{C}/\text{cm}^2$, with four samples per dose to evaluate thickness variations. As shown in **Figure S35**, we measured the thicknesses of these patches by AFM. The SU-8 thickness increases rapidly with the dose increasing and then gradually slows as the dose is large enough. The thickness deviation among samples with the same dose is less than 3 nm, indicating that precise thickness control can be achieved by adjusting the exposure dose, enabling the fabrication of subtle thickness variations and gradients.

“22. The thickness controlling experiment through EBL dose-modulation patterning.

We fabricated SU-8 samples at lower exposure doses of 2.0, 2.5, 3.0, and 3.5 $\mu\text{C}/\text{cm}^2$, with four samples per dose to evaluate thickness variations. As shown in Figure S34, the SU-8 thickness

initially increases rapidly with dose and then gradually slows as the dose continues to rise. The thickness deviation among samples with the same dose is less than 3 nm, indicating that precise thickness control can be achieved by adjusting the exposure dose, enabling the fabrication of subtle thickness variations and gradients.

Figure S34. The relationship curve between the SU-8 films thickness and the exposure dose.”

In Figure 4, we opted for PECVD to fabricate the sensing chip primarily because it is a lithography-free, scalable method that enables large-area fabrication with ease. Unlike microstructured surfaces, PECVD produces a continuous thickness gradient in the silica film rather than discrete microstructures. Thus, we provide thickness measurements rather than nanostructure characterization. To further support the reliability and repeatability of this approach, we have included additional experimental validation in **Supporting Information (Section 20)**, as shown in **response to comment 9**.

Regarding the humidity response in Figure 5, our intention was to showcase an early-stage application of the portable sensing system, captured directly using a smartphone for real-world adaptability. Given the simplicity of our humidity chamber setup and the usage of the smartphone camera, precise humidity control was not feasible, and the data points in Figures 5(iii) and 5(iv) were randomly selected for illustrative purposes. However, despite these limitations, Figure 5 clearly shows a monotonic shift in the resonance fringes with humidity variation, confirming a consistent response rather than random fluctuations. For further work, one may utilize a gas chamber to precisely calibrate both the humidity and the response of our portable sensing system, and obtain a relationship between humidity and the stripe shift of the

sensor. This process is the same as what we did in Figure 3 for refractive index sensing, as humidity sensing is also a type of refractive index sensing. Nevertheless, precise calibration of the portable system is beyond the scope of this paper.

Comment 12: I recommend that the authors address these concerns by providing more comprehensive details on the fabrication process, including any techniques employed to ensure height control, as well as supporting data and microstructural analysis to substantiate the reliability and reproducibility of the approach.

Reply and modifications: Thank you so much for your valuable suggestion.

We have revised the manuscript to provide a more detailed description of the fabrication process in the **Supporting Information (Section 21)**, including the techniques used to control the thickness of the structures, which is as follows.

“21. The detailed fabrication process for the sensor chip

In Figure 3, we controlled the SU-8 thickness by adjusting the exposure dose, with the fabrication process outlined in Figure S32.

Figure S32. *The fabrication of sensor chip based on SU-8 exposure dose controlling in Figure 3. Squares A, B and C means the square film with different exposure dose.*

For the sensor chip in Figure 4, we leveraged the inherent deposition variations of PECVD to create a continuous thickness gradient, enabling a lithography-free approach for fabricating natural thickness variations. The detailed fabrication process is shown in Figure S33.

Figure S33. *The fabrication of sensor chip based on PECVD deposition in Figure 4.”*

Specifically, in Figure 3, we employed a random dose modulation strategy in EBL, where the SU-8 thickness after development is correlated with the exposure dose. Regarding the PECVD-based fabrication in Figure 4, we emphasize that this method enables the deposition of a continuous thickness gradient rather than discrete microstructures. We have now incorporated additional microstructural characterizations and a thorough reproducibility analysis in **Supporting Information (Section 20)**, which is mentioned before, confirming that the thickness variation remains consistent across multiple fabricated samples, with a typical standard deviation of less than 2.4%. We also give some experimental evidence to illustrate the precise thickness controlling through modulating the exposure dose, which is also another way to obtain the small thickness difference/thickness gradient. We have also added this part in the **Supporting Information (Section 22)**.

These additions strengthen the reliability of our fabrication approach and provide clearer insights into height control, ensuring the reproducibility of our sensor platform. We hope this revision adequately addresses your concerns, and we appreciate your constructive feedback.

Comment 13: Furthermore, a more thorough examination of the relationship between image changes and sensitivity, particularly in response to humidity variations, is necessary to validate the reported sensitivity values.

Reply and modifications: Thank you for your valuable comments.

Regarding the humidity response in Figure 5, our intention was to showcase an early-stage application of the portable sensing system, captured directly using a smartphone for real-world adaptability. For this reason, we primarily focus on the portable sensing application, so we have not conducted further in-depth data analysis. Besides, we have conducted the refractive index-

based experiment in Figure 4, as the humidity change is same kind case of refractive index change. So, in this portable scenario, we just show the trend of the change of resonant stripe in Figure 5 captured by the hand-held camera of smartphone, instead of the in-depth data analysis. Moreover, given the simplicity of our humidity chamber setup, precise humidity control was not feasible, and the data points in Figures 5c(i) and 5c(ii) were randomly selected for illustrative purposes. However, despite these limitations, Figure 5 clearly shows a monotonic shift in the resonance fringes with humidity variation, confirming a consistent response rather than random fluctuations. Furthermore, in the future, we can do the image process through the software in smartphone, or change it to the precise humidity measurement case (such as in the fixed optical configuration).

Comment 14: Based on the data presented, it remains unclear how the sensor achieves sensitivity and range without apparent trade-offs, particularly given that the system seems to only detect small refractive index changes. Could you elaborate on how the system manages to maintain high sensitivity over a broad range of refractive index variations? Furthermore, do larger refractive index changes pose any challenges to the sensor's detection capabilities?

Reply and modifications: Thank you for your insightful comments.

In our design, the measurement range can be dynamically adjusted by reconfiguring the incident angle, enabling flexible operation without requiring increasing the number of CCD pixels. As shown in Figure 4b, when the concentration increases to 1.0%, the ring pattern almost disappears. However, by adjusting the incident angle, we can create new measurable cycles, as illustrated in Figure 4e, thereby extending the refractive index measurement range. Thus, by simply adjusting the incident angle, the new measurable cycle can widen the dynamic concentration range in an adjustable way. As shown in Figure 4e, the dynamic range ($\Delta n = \Delta n_0 + \Delta n_1 + \Delta n_2 + \dots$) can be expanded in multiples of the standard dynamic range (Δn_0), in a continuous angle re-adjusting process, mitigating the tradeoff between the sensitivity and dynamic range.

We think it's not suitable to describe it as "without" sensitivity-range tradeoff. Hence, we have revised the manuscript to better articulate this point and avoid the impression that the sensitivity-range tradeoff is entirely eliminated. In the revised manuscript, we have emphasized

that our system provides a flexible means to overcome or mitigate this tradeoff through reconfiguration, as demonstrated in the manuscript.

We have also removed the word “tradeoff” in the title and changed the title of the manuscript to “*Ultrasensitive Imaging-based Sensor Unlocked by Differential Guided-Mode Resonance*”.

Comment 15: In Figure 5(ii), the non-specific adsorption curve for R-IgG shows only a 3-pixel shift, which seems inconsistent with the claim of "ultrasensitivity" made in the manuscript. This raises concerns about the validity of the sensitivity calculation method employed. Could you provide a more thorough explanation and justification for the sensitivity calculation, especially considering the discrepancy between the expected and observed shifts? Additionally, in light of the ultra-sensitivity claimed, does the sensor demonstrate robust performance in practical applications, such as the detection of ultra-low concentration gases or even single-molecule detection? As a reviewer, I recommend that the authors offer more detailed experimental evidence to substantiate these claims. Specifically, it would be valuable to include data demonstrating the sensors' real-world performance in detecting very low concentrations of gases or single molecules. Furthermore, comparisons with other established sensing methods or devices would be beneficial to validate the performance and highlight the advantages of the proposed sensor.

Reply and modifications: Thank you for your insightful comment.

The previous results in Figure 5(ii) may not be so impressive due to low efficiency of the surface modification and binding. To improve the sensitivity performance in Figure 5(ii), we have carried out a new experiment and adopted an optimized modification strategy and used streptavidin-biotin, known for its strong affinity, as the detection target (Biosens. Bioelectron. 2015, 67: 230-236; Phys. Chem. B 2009, 113, 25, 8776-8783; Anal. Chem. 2010, 82, 12, 5211-5218). The new result is added in Figure 5. The new experimental result shows a more evident pixel shift ~20 pixels, and the theoretical detection limit can be estimated as small as 15 pM.

We conducted real-time monitoring of the 1 nmol/L biotin binding process, with the resulting resonance stripe shift shown below (**Figure S23**). The stripe moves fast to the left at the beginning, that means the increasing refractive index of the surface. Then the stripe moves gradually as the time increases.

Figure S23. Real-time monitoring of the 1 nmol/L biotin binding process. The red dashed line represents the movement of the stripes, while the white dashed line represents the background that does not move for reference.

As shown in **Figure S24**, the stripe curve of image gray shows a clear shift as the time increases during the binding process.

Figure S24. The stripe curve of image gray during the binding process. The arrow in the figure means the direction of time increasing, following with the peak shift.

The pixel shift curve for 1 nmol/L biotin is presented in the **Figure S25**. A significant shift of over 20 pixels was observed at this low concentration. The binding signal increased rapidly at the initial stage and gradually reached saturation over time.

Figure S25. The pixel shift curve for 1 nmol/L biotin binding process.

Based on the calculated pixel shift standard deviation (~ 0.3 pixels), which is mentioned in the **Supporting Information (Section 11.3)**, the theoretical detection limit can be estimated as low as 15 pM ($1 \text{ nM}/(20/0.3)$).

Correspondingly, we have revised **Figure 5** and added a corresponding description (Page 14, above Figure 5) as follows:

“

Figure 5. The surface sensing and monitoring and the portable device designed for this thickness-modulated refractive index sensor. **(a)** The monitoring response curve of the biotin molecule binding event (the concentration is 1 nmol/L). The insert is the surface modification process. APTES means 3-aminopropyltriethoxysilane. GA means glutaraldehyde. SA means streptavidin. BSA means bovine serum albumin. **(b)** The portable device design and its application. (i) A portable prototype ($20 \times 14 \times 8 \text{ cm}^3$) based on a plug-and-play sensor chip. (ii) Different ring-stripe images under various solution conditions. **(c)** (i) Two-dimension humidity sensing based on this thickness-modulated chip. (ii) The ring-stripe images as the humidity increases in a chamber, captured by the smartphone camera.

“The high-specificity binding between streptavidin and biotin is a well-established model in biomolecular recognition, playing a crucial role in medical diagnostics, food safety, and environmental monitoring. As shown in Figure 5a, we functionalized the sensing surface by silanization and streptavidin incubation to enable biotin detection. The detailed surface modification steps are provided in the Supporting Information (Section 15). Even at a low biotin concentration of 1 nmol/L, our sensor exhibits a signal response exceeding 20 pixels.”

Additionally, we have added detailed modification methods, procedures, and results in the

Supporting Information (Section 15), as outlined below.

“15. The detailed modification and some results regarding the biotin test in Figure 5a

The surface modification process involved: Firstly, conducting silanization treatment to the deposited SiO₂ using 2% (v/v) APTES (3-aminopropyltriethoxysilane, C₉H₂₃NO₃Si) for 40 min. Secondly, conducting glutaraldehyde (5% (v/v)) treatment for 1 hour. Thirdly, conducting streptavidin incubation (200 µg/mL) for 1 hour. Then blocking other binding sites with 5 mg/mL BSA (bovine serum albumin) for 1 hour. Finally, monitoring the biotin solution (1 nmol/L) binding for 50 min.

We conducted real-time monitoring of the 1 nmol/L biotin binding process, with the resulting resonance stripe shift shown below (Figure S23). The stripe moves fast to the left at the beginning, that means the increasing refractive index of the surface. Then the stripe moves gradually as the time increasing.

Figure S23. Real-time monitoring of the 1 nmol/L biotin binding process. The red dashed line represents the movement of the stripes, while the white dashed line represents the background that does not move for reference.

As shown in Figure S24, the stripe curve of image gray shows a clear shift as the time increases during the binding process.

Figure S24. The stripe curve of image gray during the binding process. The arrow in the figure means the direction of time increasing, following with the peak shift.

The pixel shift curve for 1 nmol/L biotin is presented in Figure S25. A significant shift of over 20 pixels was observed at this low concentration. The binding signal increased rapidly at the initial stage and gradually reached saturation over time.

Figure S25. The pixel shift curve for 1 nmol/L biotin binding process.”

We have spent a lot of time and efforts in revising this manuscript, particularly carrying out the three new experiments that this reviewer requested. Nevertheless, we very much appreciate your suggestions and comments, which have indeed helped improve greatly the quality of our manuscript. We hope our reply and modifications can address your concerns.